# A Queueing-Theoretic Framework for Stability Analysis of LLM Inference with KV Cache Memory Constraints

## Abstract

The rapid adoption of large language models (LLMs) has created significant challenges for efficient inference at scale. Unlike traditional workloads, LLM inference is constrained by both computation and the memory overhead of key–value (KV) caching, which accelerates decoding but quickly exhausts GPU memory. In this paper, we introduce the first queueing-theoretic framework that explicitly incorporates both computation and GPU memory constraints into the analysis of LLM inference. Based on this framework, we derive rigorous stability and instability conditions that determine whether an LLM inference service can sustain incoming demand without unbounded queue growth. This result offers a powerful tool for system deployment, potentially addressing the core challenge of GPU provisioning. By combining an estimated request arrival rate with our derived stable service rate, operators can calculate the necessary cluster size to avoid both costly over-purchasing and performance-violating under-provisioning. We further validate our theoretical predictions through extensive experiments in real GPU production environments. Our results show that the predicted stability conditions are highly accurate, with deviations typically within 10%.

## 1 Introduction

Since the public release of ChatGPT in late 2022, the world has witnessed the rapid rise and widespread adoption of generative artificial intelligence (AI), particularly large language models (LLMs) (Brown et al., 2020; Chowdhery et al., 2023; OpenAI, 2023; Kaplan et al., 2020; Wei et al., 2022). These models have quickly become embedded in everyday workflows across education, business, entertainment, and research (OpenAI, 2019; GitHub, 2021; Huang et al., 2025). Recent reports (GilPress, 2024; Gordon, 2024) suggest that billions of user interactions have already taken place through LLM-powered applications. While the deployment of LLMs has substantially improved human productivity, this surge in demand also creates unprecedented challenges in reliably and efficiently serving LLM inference at scale.

At the heart of this challenge lies the process of LLM inference—the computation by which a pre-trained LLM receives input prompts and generates output tokens sequentially. Unlike traditional machine learning models, LLMs often require massive computational resources, long response times, and intricate scheduling of GPU clusters. The inherently sequential generation of tokens exacerbates these issues, as each step depends on the model's state from the previous step. Thus, efficient inference is not merely a matter of computational optimization, but also one of system-level design to balance responsiveness, throughput, and resource costs.

A central difficulty in this design problem is understanding and ensuring the stability of LLM inference under queueing dynamics. From a systems perspective, an inference service can be modeled as a queueing network where user requests arrive randomly, wait for GPU allocation, and are processed token by token. If the arrival rate of requests exceeds the system's effective service rate, queues will grow unbounded and the service will become unstable, leading to unacceptable latency and degraded user experience. Consequently, it is crucial to estimate the throughput and stability conditions of LLM inference in advance. Accurate estimation enables practitioners to (i) provision the appropriate number of GPUs to meet anticipated demand, (ii) design scheduling algorithms

that minimize latency while maximizing utilization, and (iii) develop adaptive scaling strategies that respond to fluctuating workloads in real time.

In modern LLM inference, a crucial technique for improving efficiency is the use of the key–value (KV) cache (Pope et al., 2023; Kwon et al., 2023). During the process of the LLM inference, instead of recomputing the query, key, and value representations for all previously generated tokens at every step, the model stores the key and value vectors in memory and reuses them for subsequent computations. This mechanism reduces the per-token computational cost of decoding from scaling with the sequence length to scaling only with the new token, thereby greatly accelerating inference. However, this acceleration comes at the expense of substantial GPU memory consumption, since the cache must retain the representations for all tokens in the active context. For requests with long prompts or long generations, the memory footprint can quickly become a bottleneck, often dominating system capacity constraints.

This memory–computation tradeoff introduces unique challenges for queueing-based modeling of LLM inference. Traditional queueing models focus primarily on computation as the limiting resource, but in LLM serving systems, both GPU compute throughput and memory availability jointly determine stability. Therefore, queueing models for LLM inference must explicitly account for the interplay between computation and memory constraints, requiring new formulations that go beyond classical service-rate abstractions.

In this paper, we develop a queueing-theoretic framework that explicitly incorporates KV cache and memory constraints into the analysis of LLM inference. To the best of our knowledge, this is the first queueing model that captures the memory dimension of GPU-based inference, which has emerged as a critical bottleneck in large-scale LLM serving. Our main contributions are threefold:

1. **Modeling.** We introduce a queueing-theoretic model for LLM inference that explicitly incorporates both computational demand and GPU memory constraints, the latter governed by the Key-Value (KV) cache. It generalizes the queueing LLM inference model in Li et al. (2025) by adding memory awareness and refines Jaillet et al. (2025) by reformulating its scheduling setup into a queueing framework with a chunked prefill stage. This enables a more realistic characterization of system behavior under varying prompt lengths, output lengths, and request arrival patterns.

2. **Theoretical Analysis.** Building on this model, we derive rigorous stability and instability conditions for LLM inference systems. These results provide a principled foundation for determining when a serving system can reliably handle incoming workloads, and when unbounded queue growth or system failures may occur.

3. **Empirical Validation.** We conduct extensive experiments in real GPU production environments, evaluating both single-GPU and multi-GPU configurations. Our results demonstrate that the derived stability conditions are highly accurate in practice, with prediction errors typically within 10%. These findings confirm the practical relevance of our model and highlight its potential for guiding system design and resource provisioning.

The remainder of the paper is organized as follows. Section 2 surveys the relevant literature on stochastic bin packing problems and LLM inference. In Section 3, we present our queueing model for LLM inference. The stability conditions for LLM inference under memory constraints are derived in Section 4. Section 5 presents real GPU experiments that validate our theoretical results. Finally, we conclude the paper and discuss limitations and directions for future work in Section 6.

## 2 LITERATURE REVIEW

**Queueing and LLM Inference.** Recent work has begun to view large language model (LLM) inference through the lens of queueing theory. Wu et al. (2023) address the system-level challenge of serving LLMs under strict latency requirements by introducing token-level preemptive scheduling and a memory-aware multi-level feedback queue. While their approach is inspired by queueing theory, it does not provide a precise queueing model or theoretical analysis. Complementing this, Yang et al. (2024) develop queueing-theoretic models to analyze the impact of stochastic output token lengths and batching policies, showing that strategies such as token clipping and elastic batching can be quantitatively optimized to balance throughput and delay. Furthermore, Li et al. (2025) study

throughput-optimal scheduling algorithms and stability conditions. Guldogan et al. (2024) study the multi-bin batching strategy also from a queueing perspective. However, all of these works do not explicitly model memory constraints or the KV cache, which can lead to substantial errors in scenarios where memory is a tight bottleneck. At a broader level, Mitzenmacher & Shahout (2025) survey the intersection of queueing, prediction, and LLM-specific constraints, highlighting open problems such as robust scheduling with noisy predictions, incorporating memory footprints into queueing models, and reconciling throughput–latency tradeoffs.

**Theoretical Modeling for LLM Inference.** A parallel line of theoretical work focuses on scheduling algorithms for LLM inference (Jaillet et al., 2025; Ao et al., 2025; Wang et al., 2025; Chen et al., 2025). These studies typically model LLM inference as a resource-allocation problem in Operations Research. While our goal is to characterize the fundamental stability limit (i.e., the maximum processing rate of a given system) under stochastic prompt arrivals, prior scheduling work focuses on designing policies that improve system efficiency, typically in offline settings with potentially adversarial prompt sizes, without explicitly deriving a closed-form expression.

**Stochastic Bin Packing Problems.** LLM inference with memory constraints is similar to stochastic bin packing problems, or equivalently, queueing systems with packing constraints, which have been well-studied in the literature. In stochastic bin packing problems, each request with a given bandwidth arrives online in a stochastic manner. The system manager must develop scheduling algorithms to serve the requests while satisfying the total bandwidth constraints. This problem was first modeled in Coffman & Stolyar (2001). Gamarnik (2004) then derived the stability conditions for this model under the Best-Fit scheduling policy, where the largest requests are allocated first, followed by the next largest, and so on. Subsequently, Maguluri et al. (2012) later generalized the model beyond additive bandwidth settings to more general configurations, studying the allocation of virtual machines to physical computers and showing that the Best-Fit algorithm is generally not throughput-optimal. Stolyar (2013); Stolyar & Zhong (2013; 2015) considered the infinite-server version of this problem, where the goal is to minimize the number of servers used. Furthermore, Ghaderi et al. (2014) proposed a randomized Best-Fit algorithm in this context, and Gupta & Radovanovic (2015) generalized the problem to cases where requests may renege and servers may slow down.

There are several key differences between stochastic bin packing problems and LLM inference with memory constraints. First, the memory footprint of a request is not a fixed count but evolves approximately linearly as processing progresses. Second, LLM inference consists of two distinct phases—the prompt phase and the token generation (decode) phase—with different memory dynamics. Third, the total available memory is typically much larger than the memory requirement of a single request, which naturally places the problem in a large-memory regime.

## 3 MODEL

To model the LLM inference process, we formulate a queueing and batching problem for a single computational worker (GPU) operating over the interval $[0, +\infty)$. The worker is subject to a KV cache memory constraint $M > 0$. We normalize the KV cache unit from bytes to tokens.[1]

Prompt requests arrive sequentially over time, with their arrivals following a stationary Poisson process with rate $\lambda > 0$. Let $\mathcal{I}$ represent the instance consisting of all prompt requests within the finite time horizon. Each prompt request $i$ has a size $s_i > 0$, defined as the number of tokens in the prompt. Additionally, each request $i$ is associated with a response length $o_i > 0$, representing the number of tokens in the response from autoregressive LLMs. We assume that $\{s_i, o_i\}$ is independently sampled from a joint distribution $\mathcal{F}_{\text{in-out}}$ with the joint probability mass function of the number of tokens in the prompt and the corresponding response length denoted as $p(s, o)$.

In practice, a system designer selects hardware with a memory capacity $M$ that is significantly larger than the typical request size $(s_i + o_i)$ under the distribution $\mathcal{F}_{\text{in-out}}$. This enables high concurrency by allowing multiple requests to be processed simultaneously. Therefore, we focus on the regime where

---

[1]$M$ depends on the GPU's computational power and the complexity of the large language model in use. Moreover, if we allow swapping requests from GPU to CPU, then $M$ is the total amount of memory of the given GPU and CPU in terms of tokens.

the memory requirement of a single prompt is much smaller compared with the GPU's memory capacity, i.e., $\operatorname{ess\,sup}_{s,o\sim\mathcal{F}_{\text{in-out}}}\{s+o\} \ll M$.

**Prompt Processing.** Each prompt request is processed online and undergoes two primary phases on the GPU worker:

1. *Prompt Phase:* Following a widely adopted approach Agrawal et al. (2024b); Ratner et al. (2022); Yen et al. (2024); An et al. (2024), we assume that input prompts are divided into equally sized small chunks. We assume that each chunk contains $\hat{s}$ number of tokens. Therefore, each prompt will be split into $s_i/\hat{s}$ number of chunks[2]. These chunks are processed sequentially. During this phase, the memory required for processing the $j$th chunk of prompt $i$ is $j \times \hat{s}$, meaning memory usage gradually increases as the prompt is processed. When the final chunk is processed, the total memory occupied in the KV cache for prompt $i$ reaches $s_i$. Once the prompt is fully processed, the model generates its first output token.

2. *Decode Phase:* After the prompt phase, subsequent tokens are generated sequentially. In this phase, the memory required for processing the $j$th token of prompt $i$ ($j \in [o_i]$) is $s_i+j$. This increase occurs because each new token adds its key and value to the KV cache, contributing additional usages of memory. Consequently, by the time the final token of prompt $i$ is processed, the total memory usage reaches $s_i + o_i$. After generating the last token, the KV cache memory allocated to prompt $i$ is fully released.

**Mixed Batching.** A mixed batch may contain any unprocessed prompt chunk or output token from different requests (Agrawal et al., 2023; 2024b). When a prompt chunk is added to a batch, its next chunk is generated after processing, and if it is the final chunk, the first output token is produced once the batch completes. Similarly, when an output token is included in a batch, the next token is generated upon batch completion.

Since batch processing time remains relatively stable in practice, we normalize it to one unit of time in our model. Therefore, service decisions are made at discrete time slots. Specifically, one unit of time corresponds to $\bar{b}$ seconds, where $\bar{b}$ represents the average batch processing time for a given computational worker and the large language model in use. We argue that the assumption of constant batch processing time is reasonable for stability analysis, as real-world traces show over 80% of batches exhibit identical processing times (See Figure 1 and more real traces can be found in Appendix B).

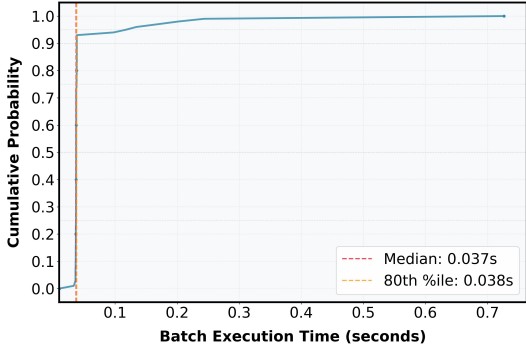

Figure 1: Cumulative Distribution Function for Batch Execution Time with PD ratio 1:1 requests

In LLM services, exceeding the KV cache limit in both the GPU and the CPU causes the system to stop working. The batching constraint ensures that the total memory usage at any time does not exceed $M$, considering all ongoing prompt requests (i.e., those still being processed or awaiting final output tokens). Formally, let $S^{(t)}$ denote the set of prompts that have been processed but still have pending output tokens at the end of time $t$. For each $i \in S^{(t)}$, define $c_i^{(t)}$ as the index of prefill chunk of request $i$ having processed at the end of time $t$, where $c_i^{(t)} \in \{1, 2, \ldots, s_i/\hat{s}\}$, and define $d_i^{(t)}$ as the index of output token of request $i$ having processed at the end of time $t$, where

---

[2]To simplify the analysis, we assume that $s_i/\hat{s}$ is a positive integer for all $i$.

$d_i^{(t)} \in \{0, 1, \ldots, o_i\}$. Therefore, if $c_i^{(t)} < s_i/\hat{s}$, the index of output token will always be zero, i.e., $d_i^{(t)} = 0$. The memory constraint requires that, for all $t \in [0, +\infty)$, at the end of period $t$, the total memory usage must satisfy

$$\sum_{i \in S^{(t)}} c_i^{(t)} \hat{s} + d_i^{(t)} \leq M.$$

**Work-conserving Scheduling and Continuous Batching.** In this work, we consider continuous batching. We also allow any work-conserving scheduling policy, meaning the server processes the next-priority prompt whenever it has available capacity. Examples include first-come-first-served (FCFS), as in vLLM (Kwon et al., 2023) with SARATHI (Agrawal et al., 2024b), and other disciplines such as Shortest Job First (SJF).

To rigorously define the work-conserving scheduling and the continuous batching in mathematics, we consider at any $t \in [0, T]$, the policy prioritizes all ongoing processing requests and then admits new prompts into the batch according to a specified priority rule. Specifically, a new prompt is added whenever doing so does not violate the memory constraint. If, at a later time, the memory constraint is violated due to the increasing KV-cache size, the system can swap the KV cache of some prompts to the CPU at no cost. This model closely approximates the actual scheduling algorithm used in vLLM.

**System Stability.** At each discrete time step, we represent the system state by the set of prompts that have arrived but are not yet completed, $S^{(t)} \cup W^{(t)}$, along with the indices of the prefill chunk and output token processed for each request $i \in S^{(t)} \cup W^{(t)}$ at the end of time $t$, denoted by $\{c_i^t, d_i^t\}$. Here, $W^{(t)}$ denotes the set of prompts that have arrived but have not yet started processing at time $t$, where $c_i^t = d_i^t = 0$ for $i \in W^{(t)}$. This induces a countable-state, discrete-time Markov chain. For simplicity, we assume that the Markov chain is irreducible. In this setting, **stability** corresponds to the **positive recurrence** of the Markov chain (Dai & Harrison, 2020; Bramson, 2008). Intuitively, stability means that the backlog of unprocessed prompts does not grow without bound.

**Stability Condition.** Since the arrival process is a stationary Poisson process with rate $\lambda$, the arrival rate of the queueing system is also $\lambda$, which is known. We define the processing rate $\mu$ as the maximum expected number of requests that can be fully processed per second under continuous utilization (when the server is never idle). By standard queueing theory, the system is **stable** if $\lambda < \mu$ and **unstable** if $\lambda \geq \mu$.

However, computing $\mu$ is highly nontrivial, as it depends on several factors:

1. *KV cache limit $M$:* A larger memory limit allows more requests to be processed simultaneously.

2. *Input size and output length joint distributions $\mathcal{F}_{in\text{-}out}$:* These distributions directly influence the processing time of each request.

3. *Scheduling and batching algorithm $\mathcal{A}$:* Different scheduling and batching strategies affect processing efficiency. In this work, we default FCFS with continuous batching, closely aligned with real-world implementations.

4. *Averaged processing time $\bar{b}$:* The expected time required to process a batch also impacts the overall processing rate.

Thus, we express $\mu$ as a function:

$$\mu = f(M, \mathcal{F}_{\text{in-out}}, \mathcal{A}, \bar{b}).$$

A key challenge in deriving a closed-form expression for $f$ arises from the dynamic memory usage during LLM inference. Initially, each request's prompt chunk consumes minimal memory, allowing many requests to be processed together. However, as processing progresses and output tokens are generated, memory usage grows linearly for each request. This causes the number of requests that can be processed simultaneously to decrease over time, leading to significant variability in batch sizes. The interplay between these factors makes it difficult to determine the exact processing rate $\mu$. In the next section, we present a closed-form expression for $f$ and provide a rigorous proof.

## 4   THE STABILITY AND INSTABILITY CONDITIONS

In this section, we give rigorous theory on the stability and instability conditions of LLM inference. First, we note that for an arriving request with an input size $s$, output length $o$, and the chunk size $\hat{s}$, the lifetime cumulative memory usage (summed over the entire service of that request) is

$$g(s,o) := (\hat{s} + 2\hat{s} + \cdots + s) + ((s+1) + (s+2) + \cdots + (s+o)) = \frac{(1 + s/\hat{s})s + 2os + (1+o)o}{2}.$$

To begin with, let us define

$$\mu = \frac{M}{\bar{b}\mathbb{E}_{s,o \sim p(s,o)}\big[g(s,o)\big]} = \frac{M}{\bar{b}\sum_{s,o} g(s,o)\, p(s,o)}. \tag{1}$$

We show the stability conditions in two steps:

1. We show that $\lambda > \mu$ implies the system is overloaded. (Theorem 1)
2. We show that $\lambda < \mu(1 - \delta)$ ensures stability, where $\delta \ll 1$ defined below. (Theorem 2).

**Theorem 1.** *If $\lambda > \mu$, then under any scheduling policy, the total number of requests in the system grows to infinity. Namely, the system is overloaded.*

Theorem 1 is relatively straightforward. If $\lambda > \mu$, the total memory requirement exceeds the maximum GPU memory capacity. The detailed proof is provided in Appendix A.

**Theorem 2.** *Let $\delta = \frac{\operatorname{ess\,sup}_{s,o \sim \mathcal{F}_{in\text{-}out}}\{s+o\}}{M} \ll 1$. If $\lambda < \mu(1 - \delta)$, then under the work-conserving scheduling and batching policy described in Section 3, the system is stable.*

*Proof Sketch.* Our stability analysis builds on a Lyapunov argument (Brémaud, 2013), adapted to the unusual memory dynamics of LLM inference. The key idea is to track the *total outstanding memory demand* $V(t)$, defined as the cumulative future KV cache usage required by all active or waiting requests at time $t$. This quantity grows as new requests arrive, and decreases as the GPU processes batches.

The challenge is that a request's KV usage is piecewise linear: it grows by chunks in the prompt phase and then by tokens in the decode phase. We overcome this by collapsing each request's entire future KV trajectory into a single *lifetime memory footprint* $g(s,o)$, and $V(t)$ behaves like "area left to erase." Per slot (of length $\bar{b}$), processing with KV utilization $U_t$ *reduces* $V$ by exactly $U_t$ (we are erasing that much area), independent of whether the consumed work comes from prompt or decode tokens. Under the work-conserving scheduling policies with lookahead and the memory slack $\delta := \operatorname{ess\,sup}_{s,o}(s+o)/M$, a large backlog implies a *saturation* property: the batcher can keep utilization $U_t \geq M(1 - \delta)$ throughout the slot without risking overflow (the $\delta$ fraction is the headroom needed to accommodate the largest single job).

Meanwhile, new arrivals in a slot contribute an *expected* increase of $\lambda \bar{b}\,\mathbb{E}[g(s,o)]$ to $V$. Hence, the conditional one-step drift satisfies

$$\mathbb{E}[V(t+1) - V(t) \mid \text{state at } t] \;\leq\; -M(1-\delta) \;+\; \lambda \bar{b}\,\mathbb{E}[g(s,o)].$$

If $\lambda \bar{b}\,\mathbb{E}[g(s,o)] < M(1-\delta)$, i.e., $\lambda < \mu(1-\delta)$, the drift is strictly negative outside a compact set. By the Foster–Lyapunov criterion, the Markov chain is positive recurrent, so the system is stable. Intuitively, once the backlog is large, the GPU continuously "erases area" at rate $M(1-\delta)$, while the stochastic inflow adds area at rate $\lambda \bar{b}\,\mathbb{E}[g(s,o)]$; the former dominates under the stated condition. □

The detailed proof is provided in Appendix A. In practice, a GPU's total memory far exceeds the memory needed for a single request. Consequently, Theorems 1 and 2 together offer a practical criterion for assessing the stability of LLM inference.

If we have reliable estimates of the arrival rate $\lambda$ and the service rate $\mu$, this information can not only provide the stability condition for a single-GPU setting, but also guide the flexible provisioning of multiple GPUs. For instance, if system managers aim for a target utilization rate of $\rho = 90\%$,

they can estimate the required number of GPUs as approximately $\lambda/(\mu \cdot \rho)$. This approach enables efficient resource allocation, ensuring that the system maintains high throughput while avoiding excessive queuing or memory bottlenecks. Moreover, our results with dynamic estimates of $\lambda$ and $\mu$ can support dynamic scaling strategies, where GPUs are added or removed in response to fluctuating workloads, further improving operational efficiency.

## 5 NUMERICAL EXPERIMENTS

In this section, we conduct numerical experiments to validate the accuracy of our theoretical stability condition against real-world measurements. Section 5.1 describes the details of our implementation and hardware environment. Section 5.2 verifies our theory in the single-GPU setting, showing that the error remains within 10%. Section 5.3 demonstrates the robustness and generalizability of our theory in multi-GPU settings, where it still achieves near-perfect predictive power.

### 5.1 IMPLEMENTATION AND HARDWARE DETAILS

**Testbed.** Our hardware setup uses eight identical NVIDIA A100 GPUs, with one GPU per replica; we disable intra-model parallelism (tensor parallel size = 1, pipeline stages = 1), so all concurrency comes from replica-level data parallelism across the 8 GPUs. The served LLM is Meta-Llama-3-8B. Each replica runs the vLLM v1 (Kwon et al., 2023) inference engine with chunked prefill enabled: prompts are split into fixed-size segments to bound per-step memory use and interleave admission of other requests.

**Workload Distributions.** We evaluate our stable condition under three distinct prefill-decode (PD) ratio regimes by defining different joint distributions $\mathcal{F}_{\text{in-out}}$ for input (prefill) and output (decode) token lengths:

1. **PD Ratio 1:1:** Prefill (input length) is independently sampled from Uniform(10, 1600), and decode (output lengths) of each request equals to its prefill value.

2. **PD Ratio 2:1:** Prefill is independently sampled from Uniform(10, 2133), and decode (output lengths) of each request equals to $0.5\times$ its prefill value.

3. **PD Ratio 1:2:** Prefill is independently sampled from Uniform(10, 2133), and decode (output lengths) of each request equals to $2\times$ its prefill value.

Here, we control the decode length per request using the functionality in vLLM v1, which allows setting upper and lower bounds. For instance, in the 1:1 PD ratio setup, we sample a prefill length and set both the lower and upper decoding bounds to this value, thereby enforcing the desired ratio.

**Parameter Selection.** The memory capacity parameter $M$ is set to 131,000 tokens, based on the observed maximum capacity of the combined GPU and CPU-swapped memory in our testbed.

To determine the characteristic processing time $\bar{b}$, we adopted a data-driven approach. Empirical analysis of real traces Figure 5 in Appendix B revealed that the median batch processing time is a robust estimator, aligning closely with the 90th percentile in many cases. For each experimental setup, we simulated 10,000 jobs from the respective distribution to form a training dataset. We then computed the median processing time of these batches to obtain $\bar{b}$ for that specific workload:

- **PD Ratio 1:1:** $\bar{b} = 0.0372$s; **PD Ratio 2:1:** $\bar{b} = 0.0430$s; **PD Ratio 1:2:** $\bar{b} = 0.0337$s.

### 5.2 SINGLE-GPU RESULTS

We evaluate the accuracy of our theoretical stable processing rate, $\mu_{\text{theory}}$ (from Equation equation 1), against empirically measured values from the GPU, $\mu_{\text{gpu}}$. To ensure our measurement captures the system's steady-state behavior, we calculate the empirical processing rate $\mu_{\text{gpu}}$ by excluding the first and last 1,000 requests, which may be affected by initial warm-up and termination phases. The measurement time interval is defined from the arrival time of the 1,001st request to one of the last 1,000th request to make sure that requests that completed during the termination phases are not included as by including these requests, the latency statistics can be artificially improved.

The empirical rate is then computed as the total number of processed requests within this interval divided by its duration. Therefore, $\mu_{\text{gpu}}$ can be interpreted as the averaged number of requests can be completed by the real GPU within 1 second. To quantify the accuracy of our model, we use the Gap Absolute Percentage (GAP):

$$\text{GAP} = \frac{|\mu_{\text{theory}} - \mu_{\text{gpu}}|}{\mu_{\text{gpu}}}.$$

Table 1: Stable Condition of Single GPU under Different PD Ratios.

| P/D Ratio | $\mu_{\text{gpu}}$ | $\mu_{\text{theory}}$ | GAP |
|:---:|:---:|:---:|:---:|
| $1:1$ | 3.387 | 3.226 | 4.75% |
| $2:1$ | 3.650 | 4.015 | 10.00% |
| $1:2$ | 2.969 | 2.926 | 1.45% |

In Table 1, all values converge when the arriving rate $\lambda \geq 5$. As shown in the table, the theoretical processing rates derived from Equation equation 1 closely align with the empirically measured rates from the GPU hardware. Across all three PD ratio settings (1:1, 2:1, and 1:2), the GAP remains below 10%, demonstrating the high accuracy of our model. This result is practically significant, as it enables reliable estimation of the required number of GPUs during system deployment to ensure stability under expected workloads.

We now examine the system dynamics under a 1:1 P/D ratio by varying the arrival rate $\lambda$. Figure 4b plots the evolution of queue length over time for arrival rates of $\lambda = 5, 20$, and $50$ requests per second. The queue length increases approximately linearly in all cases, indicating the system overload where the arrival rate exceeds the service rate. For arrival rates of $\lambda = 1, 3$, the queue length is usually upper bounded by $5$, indicating the system is stable where the service rate exceeds the arrival rate.

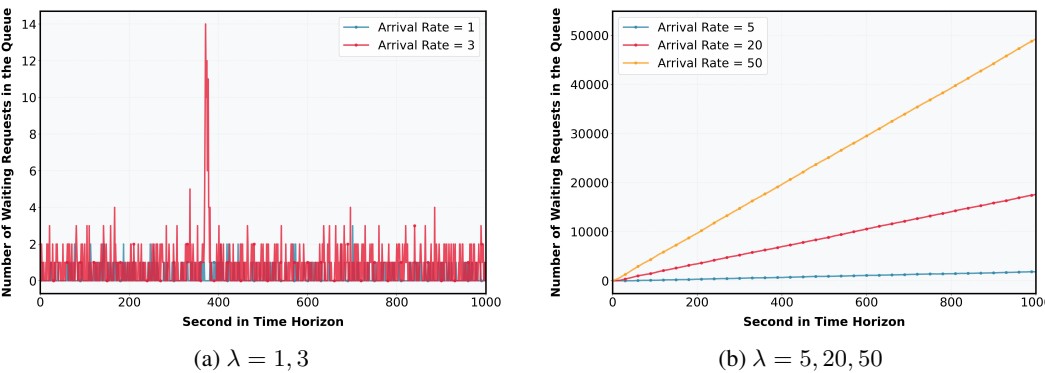

(a) $\lambda = 1, 3$         (b) $\lambda = 5, 20, 50$

Figure 2: Number of waiting requests in the queue under different arrival rates.

Furthermore, we plot the CDF of the per-request waiting time (i.e., the delay between arrival and the start of processing) in Appendix B, Figure 6, and the findings are consistent to Figure 2.

**Real Dataset Experiment** To validate our findings beyond simulations with fixed prefill/decode (P/D) ratios, we evaluate our model on the LongBench v2 dataset (Bai et al., 2025), which features highly variable and dependent P/D lengths. This benchmark contains 503 complex, long-context questions, providing a realistic distribution of request complexities. Figure 3 visualizes this relationship, showing the marginal probability densities of prefill (s) and decode (o) lengths, alongside their joint cumulative distribution function. The dataset is publicly accessible at https://longbench2.github.io/.

In our experiment setup, We partition the dataset randomly, using 80% to compute the joint probability density function $p(s, o)$ and to estimate the mean batch processing time $\bar{b}$ via a 10% trimmed

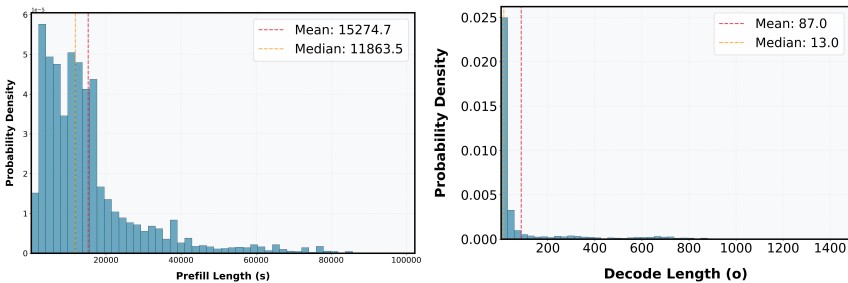

(a) Marginal probability density function of prefill length $(s)$

(b) Marginal probability density function of decode length $(o)$

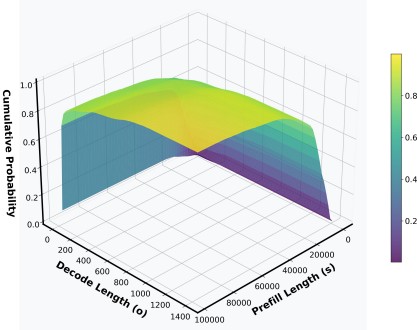

(c) Joint cumulative density function of prefill length and decode length $(s, o)$

Figure 3: Joint pdf and cdf of the LongBench v2 dataset.

Table 2: Stable Condition of Single GPU under the LongBench v2 dataset.

| $\mu_{\text{gpu}}$ | $\mu_{\text{theory}}$ | **GAP** |
|---|---|---|
| 0.610 | 0.561 | 8.03% |

average (to mitigate the effect of heavy-tailed outliers). The remaining 20% is used for testing under a load of 20 queries per second (qps).

As shown in Table 2, the theoretical stable condition $\mu_{\text{theory}}$ closely matches the empirically observed condition $\mu_{\text{gpu}}$, with an error of only 8.03%. This strong agreement underscores the importance of modeling the full joint distribution $p(s, o)$ rather than relying on simplified expectations, as it effectively captures the complex dependencies and high variance between prefill and decode lengths inherent in real-world workloads.

## 5.3 EIGHT-GPU RESULTS

Next, we conduct 8-GPU experiments to compare the real processing rate and the theoretical rate multiplies by 8, i.e. $8\mu_{\text{theory}}$, to see whether our approximation still is accurate or not in the cluster deployment. In our 8-GPU configuration, we deploy eight identical single-GPU replicas (NVIDIA A100) serving Meta-Llama-3-8B with no intra-model parallelism. Incoming requests are statelessly load-balanced across replicas using the round robin over $56,000$ requests. The PD ratio is $1:1$ and again the prefill and decode are independently sampled from Uniform(10,1600).

Table 3: Stable Condition of 8 Parallel GPUs.

| **P/D Ratio** | $\mu_{\text{gpu}}$ | $\mu_{\text{theory}}$ | **GAP** |
|---|---|---|---|
| $1:1$ | 26.710 | 25.808 | 3.38% |

As shown in Table 4, our theoretical model achieves strong convergence with empirical results for arrival rates $\lambda \geq 40$, corresponding to a per-GPU rate of $\geq 5$ requests per second. The gap between the empirical processing rate $\mu_{\text{gpu}}$ and the theoretical rate $\mu_{\text{theory}}$ is only 3.38%, demonstrating the model's high accuracy. This validates the practical utility of our approach for capacity planning: it provides an accurate lower bound on the number of GPUs required to ensure system stability—defined as a regime where no GPU is idle and the queue length remains bounded—for a given workload distribution.

Again, in Figure 4, we visualize the total queue length of 8 GPUs during processing time horizon. The insights is similar to the ones for single GPU in Figure 2: while the rates $\lambda = 40, 160, 400$ are greater than the service rate $\mu_{gpu} = 26.710$, the queue length linearly grows and the system is overloaded. For the rates $\lambda = 8, 24$, which are smaller than the service rate, the queue length is upper bounded by 13, indicating the system is stable.

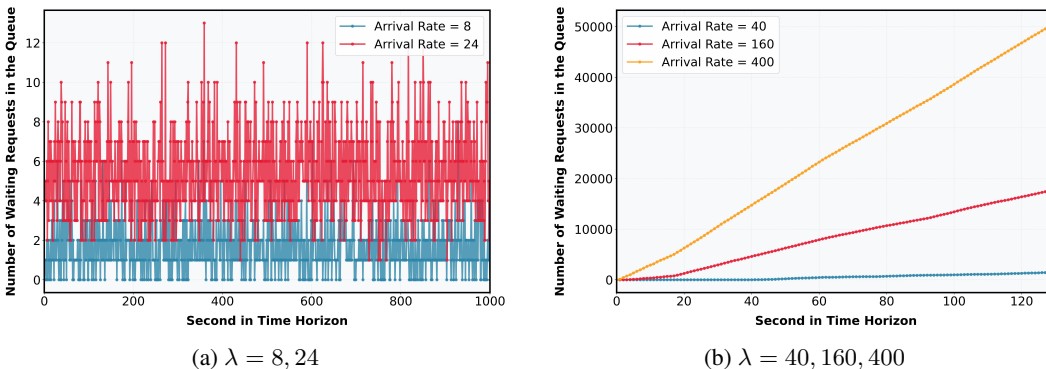

(a) $\lambda = 8, 24$  (b) $\lambda = 40, 160, 400$

Figure 4: Number of total waiting requests among 8 GPUs in the queue under different arrival rates.

## 6 CONCLUSION

This paper presents a queueing-theoretic model for LLM inference, characterizing stable GPU operation by accounting for computational demand and KV cache memory constraints. We derived the stability condition and critical processing rate $\mu$ for a single GPU, validated empirically on a real GPU testbed with less than 10% discrepancy between theoretical and measured service rates across various workloads. This provides a foundation for capacity planning, enabling system designers to estimate the minimum number of GPUs, $\lceil \lambda/\mu \rceil$, using the request arrival rate $\lambda$ and per-GPU service rate $\mu$, ensuring efficient resource provisioning to avoid over-provisioning (idle GPUs) or under-provisioning (missed service-level objectives).

Our work has limitations that suggest promising future research directions. The main theoretical-practical gap (up to 10% in the worst case) arises from model simplifications.

**1. Non-Constant Batch Processing Time.** We approximated batch processing time as a constant, using the median of empirical traces. In reality, it is a more complex function of batch composition. For example, one model in the Vidur simulator Agrawal et al. (2024a) uses an affine function to approximate service time. A promising direction for future work is to develop accurate yet tractable approximations for the stability condition under more sophisticated service time models.

**2. Request Swapping Overhead.** Our model uses a single parameter $M$ to represent the total available GPU and CPU-swapped memory. This abstraction holds well when swapping is infrequent. However, if requests are frequently swapped between GPU and CPU memory, the associated I/O overhead can become non-negligible. Extending the model to incorporate the latency cost of swapping is an important next step for environments with tight memory constraints.

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

## A    PROOFS OF THEOREM 1 AND 2

*Proof of Theorems 1.* We will show that the total outstanding memory needed in the GPU(including the chunks in the prompt phase) diverges to infinity.

For each arriving request with an input size $s$, output length $o$, and the chunk size $\hat{s}$, the cumulative memory usage in the GPU (summed over the entire service of that request) is

$$g(s, o) = (\hat{s} + 2\hat{s} + \cdots + s) + ((s + 1) + (s + 2) + \cdots + (s + o)) = \frac{(1 + s/\hat{s})s + 2os + (1 + o)o}{2}.$$

As the p.m.f. of joint distribution of arrival is $p(s, o)$, than the expected cumulative memory usage is

$$\mathbb{E}_{s,o \sim p(s,o)}\left[\frac{(1+s/\hat{s})s+2os+(1+o)o}{2}\right]$$

Given the arrival process is Poisson($\lambda$), the expected total memory demand over time period $[0, T]$ is

$$\lambda T \mathbb{E}_{s,o \sim p(s,o)}\left[\frac{(1+s/\hat{s})s+2os+(1+o)o}{2}\right].$$

Now, we turn to the service side. As each batch takes $\bar{b}$ seconds, we can take $\frac{1}{\bar{b}}$ batches per second. Moreover, at any instant in time, the memory usage cannot exceed $M$. This implies that the maximum total memory supply over the time period $[0, T]$ is:

$$T \times M \text{ (units of memory)} \times \frac{1}{\bar{b}} \text{ (batches/second)} = \frac{TM}{\bar{b}}.$$

Therefore, if $\lambda > \mu$, we have

$$\lambda T \mathbb{E}_{s,o \sim p(s,o)}\left[\frac{(1+s/\hat{s})s+2os+(1+o)o}{2}\right] - \frac{TM}{\bar{b}} \to +\infty, \text{ as } T \to +\infty$$

Hence, the total number of unprocessed tokens diverges to infinity.

$\square$

*Proof of Theorem 2.* We use a Lyapunov argument to show that, for large values of the Lyapunov function, the expected net change in the function is negative whenever $\lambda < \mu(1 - \delta)$. By the Foster–Lyapunov theorem (Brémaud, 2013, Section 5.1), this implies positive recurrence, and hence the system is stable. In the following derivation, one step corresponds to a duration of $\bar{b}$ time.

Let $V(t)$ denote the total outstanding memory demand (including the chunks in the prompt phase) in the system at time $t$. Concretely, if a request is partially completed, only its remaining memory demand contributes to $V(t)$. For a prompt $i \in S^{(t)}$ with the index of the prefill chunk $c_i^t > 0$, the remaining memory demand is given by

$$v_i(t) := \begin{cases} \left(c_i^t \hat{s} + (c_i^t + 1)\hat{s} + \ldots + s_i + (s_i + 1)\right) + (s_i + 2) + \ldots + (s_i + o_i), & \text{if } c_i^t < s_i/\hat{s}, \\ (s_i + d_i^t) + \ldots + (s_i + o_i), & \text{if } c_i^t \geq s_i/\hat{s}, \end{cases}$$

which can be simplified to

$$v_i(t) := \begin{cases} \frac{1}{2}(c_i^t \hat{s} + s)\left(s/\hat{s} - c_i^t + 1\right) + o_i s_i + \frac{(1+o_i)o_i}{2}, & \text{if } c_i^t < s_i/\hat{s}, \\ s_i\left(o_i - d_i^t + 1\right) + \frac{(o_i - d_i^t + 1)(o_i + d_i^t)}{2}, & \text{if } c_i^t \geq s_i/\hat{s}. \end{cases}$$

On the other hand, if a request has arrived but is still waiting (i.e., not yet started), namely $i \in W^{(t)}$, then its outstanding memory demand is

$$v_i(t) := g(s_i, o_i) = \frac{(1+s_i/\hat{s})s_i + 2o_i s_i + (1+o_i)o_i}{2}.$$

If a prompt has finished, it no longer contributes. Therefore, the total outstanding memory demand is

$$V(t) = \sum_{i \in S^{(t)} \cup W^{(t)}} v_i(t).$$

We now show that $V(t)$ is a valid Lyapunov function. Specifically, consider a sufficiently large constant

$$K = \sup_{s,o \sim p(s,o)} g(s,o)M.$$

We will establish that

$$\mathbb{E}\big[V(t+1)|S^{(t)} \cup W^{(t)}; \{\{c_i(t), d_i(t)\}, i \in S^{(t)} \cup W^{(t)}\}\big] \leq V(t) - \varepsilon, \quad \text{whenever } V(t) > K, \quad (2)$$

for some $\varepsilon > 0$.

We first decompose $V(t+1)$. Between time $t$ and $t+1$, some fraction of the old $W(t)$ (the tokens present at time $t$) is completed, and a random number of new requests arrive. Hence, we can write:

$$V(t+1) = \underbrace{V(t) - (\# \text{ old memory fulfilled})}_{\text{old backlog that remains}} + (\text{memory demand from newly arrived requests}).$$

$$(3)$$

If $V(t) > (\sup_{s,o \sim p(s,o)} g(s,o))M$, there are at least $M$ requests in $S^{(t)} \cup W^{(t)}$. Therefore, there are enough prompts for the GPU to process, namely,

$$(\# \text{ old memory fulfilled}) > M(1 - \delta).$$

On the other hand, we have

$$\mathbb{E}[\text{memory demand from newly arrived requests}] = \lambda \bar{b} \mathbb{E}_{s,0 \sim p(s,0)}[g(s,o)].$$

Therefore, we have

$$\mathbb{E}\big[V(t+1)|S^{(t)} \cup W^{(t)}; \{a_i(t), i \in S^{(t)} \cup W^{(t)}\}\big] < V(t) - M(1 - \delta) + \lambda \bar{b} \mathbb{E}_{s,0 \sim p(s,o)}[g(s,o)]$$
$$= V(t) + (\lambda - \mu(1 - \delta))\bar{b} \mathbb{E}_{s,0 \sim p(s,o)}[g(s,o)].$$

Due to $\lambda < \mu(1 - \delta)$ and $\mathbb{E}_{s,0 \sim p(s,o)}[g(s,o)] > 0$, we can choose

$$\varepsilon = (\lambda - \mu(1 - \delta))\bar{b} \mathbb{E}_{s,0 \sim p(s,o)}[g(s,o)],$$

which proves (2).

$\square$

# B ADDITIONAL NUMERICAL PLOTS

## B.1 BATCH PROCESSING TIME PLOT IN REAL TRACES

Figure 5 presents the cumulative distribution function (CDF) of batch execution times across several real-world traces. The subfigures are organized by workload: the first row corresponds to a prefill-decode (PD) ratio of 2:1, and the second row to a ratio of 4:1. The first column represents a low arrival rate (5 requests/second), while the second column represents a higher rate (20 requests/second). For context, Figure 1 in the main text shows the case for a 1:1 PD ratio at 5 requests/second. We observe that the batch execution time is nearly identical for different arrival rates $\lambda$. However, as the PD ratio increases, we note a corresponding increase in the disparity of execution times. This variability presents a promising direction for future modeling efforts.

## B.2 CDF FOR WAITING TIME OF EACH REQUEST

The near-linear growth of the CDF under $\lambda = 4, 5$ indicates an overloaded system. This provides further evidence for the system instability at this arrival rate and explains why the empirical processing rate $\mu_{\text{gpu}}$ converges for $\lambda \geq 5$. Moreover, under $\lambda = 2$, we can find the system is stable, and almost no request waits in the queue. Finally, for a special case $\lambda = 3.387 = \mu_{\text{gpu}}$, which is called as an unstable system in queueing theory, the growth of CDF indicates a linear trend but a non-smooth shape. This is also consistent with the theory of queueing for the unstable system.

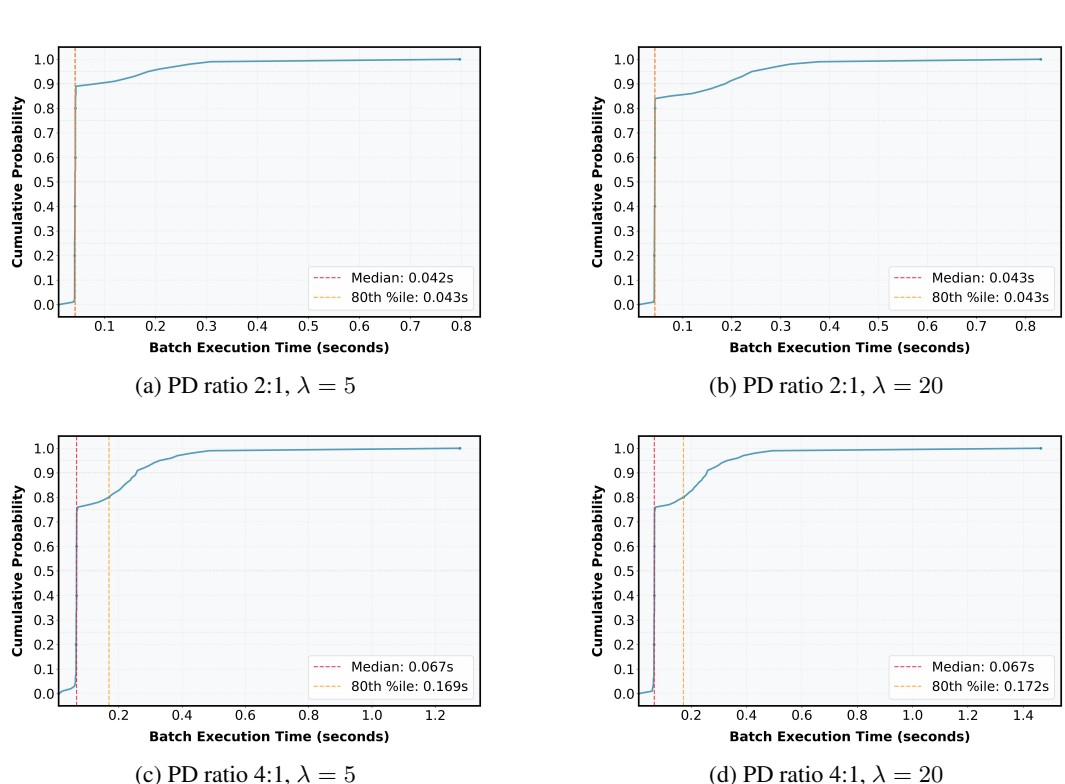

Figure 5: Cumulative Distribution Function (CDF) for Batch Execution Time under different PD ratios and arrival rates.

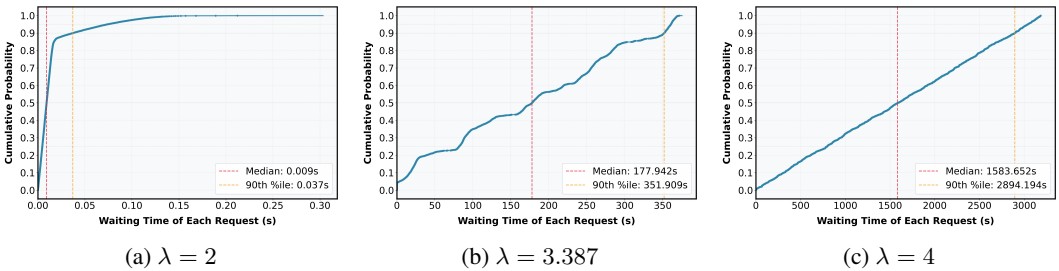

Figure 6: Cumulative Distribution Function (CDF) for Waiting Time of Each Request.

## C  STABLE CONDITION UNDER AN 8:1 P/D RATIO

This section presents a complementary numerical experiment under a more extreme workload compared to those in Section 5.2. While Section 5.2 evaluated P/D ratios of 1:1, 2:1, and 1:2—where batch processing times were largely consistent—we observe that as the P/D ratio increases, the tail of the processing time distribution becomes significantly heavier, as shown in Figure 7.

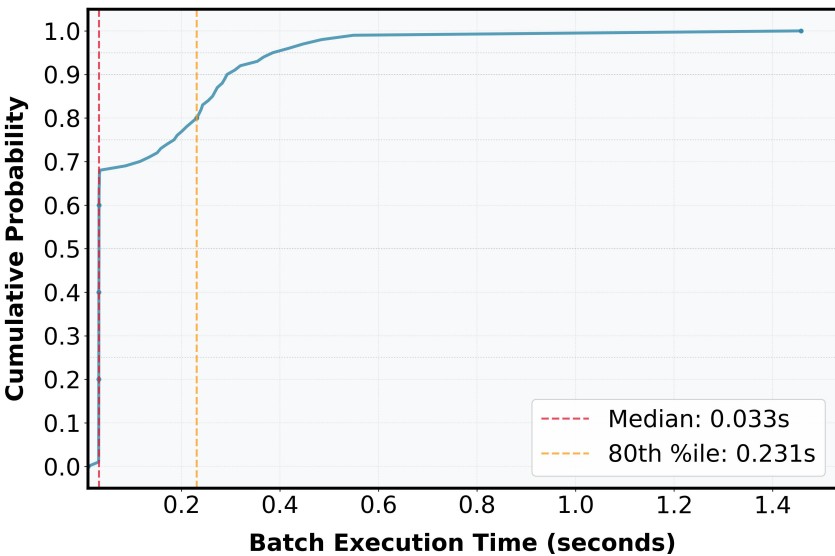

Figure 7: Cumulative Distribution Function (CDF) for Batch Execution Time under an 8:1 P/D ratio.

Figure 7 shows that while the first 70% of batches have nearly identical processing times, the remaining 30% exhibit a heavy tail. In this regime, neither the median nor the mean is a robust estimator for $\bar{b}$: the median ignores the heavy tail, while the mean is dominated by it.

A standard statistical approach for heavy-tailed distributions is the **trimmed mean**, which discards a top percentage $x\%$ of the data (treating them as outliers) and calculates the mean of the remainder. Common choices for $x$ are 5 or 10. Figure 8 shows the CDF of the trimmed batch processing time. Table 4 compares the results using these two trimmed-mean estimators for $\bar{b}$.

Table 4: Stable Condition of a Single GPU under an 8:1 P/D Ratio.

| $x\%$ **Trimmed Mean** | $\mu_{\text{gpu}}$ | $\mu_{\text{theory}}$ | **GAP** |
|---|---|---|---|
| 5% | 5.470 | 4.977 | 9.0% |
| 10% | 5.470 | 5.862 | 7.2% |

The results in Table 4 demonstrate that even under the extreme heavy-tailed distribution of an 8:1 P/D ratio, our theoretical model remains accurate. Using a robust estimator for $\bar{b}$ yields a predicted service rate $\mu_{\text{theory}}$ that is close to the empirically measured $\mu_{\text{gpu}}$, with gaps of only 9.0% and 7.2%. This highlights the robustness of our theoretical approach when combined with appropriate statistical estimation techniques.

## D  LLM USAGE

We use LLM purely for polishing writing.

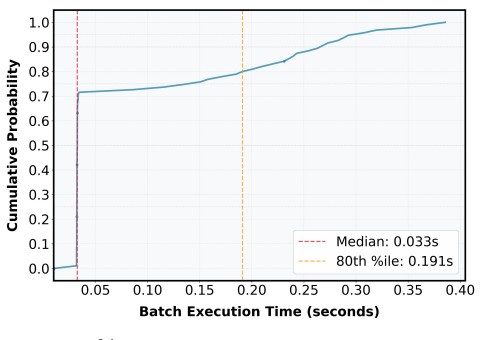 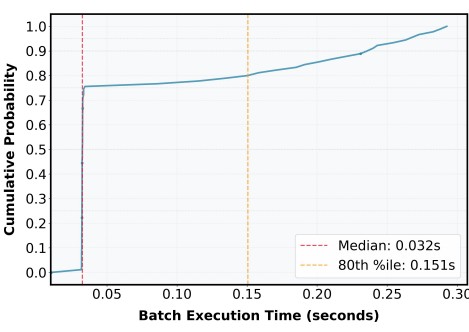

(a) 5% Trimmed Batch Execution Time

(b) 10% Trimmed Batch Execution Time

Figure 8: Trimmed Cumulative Distribution Function (CDF) for Batch Execution Time.

