# OpenReview forum: "A Queueing-Theoretic Framework for Stability Analysis of LLM Inference with KV Cache Memory Constraints"
_ICLR.cc/2026/Conference — Submitted to ICLR 2026_

### Official Review · Reviewer_86fz · 2025-10-24

**Soundness:** 4
**Presentation:** 3
**Contribution:** 2
**Rating:** 4
**Confidence:** 5

**Summary:**

Serving LLMs is an important and timely problem. One of the critical questions systems engineers face is determining the maximum load a system can serve while maintaining reasonable per-request waiting times. In practice, this is often verified empirically by incrementally increasing the workload and identifying the "inflation point" on the goodput curve. The authors propose a more theoretically-grounded method to determine this inflation point, beyond which the system becomes unstable. The authors model this behavior for a system using FCFS scheduling with a chunked prefill batching strategy, deriving a closed-form expression for the stability condition. Their experiments show that their model achieves high accuracy, with a reported error rate of up to 10%.

**Strengths:**

1. The authors target an extremely important and practical problem. From a systems designer's perspective, having a closed-form solution for system stability is a valuable tool that could significantly inform autoscaling strategies.

2. The choice to model available memory using the number of fittable KV cache tokens, rather than raw memory capacity, is really nice. This abstraction simplifies the analysis and makes the problem formulation more intuitive.

3. Despite a few minor grammatical issues, the paper is well-written and enjoyable to read. It presents a clear narrative, making the work's value easy to understand.

**Weaknesses:**

My primary concern stems from the assumption on line 184 (and Figure 1) that "batch processing time remains relatively stable in practice." I find this claim questionable, as it forms the foundation for the rest of the analysis.

While the batch processing time of LLMs is _predictable_, this does not mean it is _stable_. Figure 1 itself illustrates this point: although many requests (~90%) have a runtime near 40 ms, the distribution has a significant long tail, with runtimes extending to 700 ms. This long tail, which is characteristic of LLM serving, contradicts the notion of a stable runtime.

This runtime variance is well-understood to originate from the computationally expensive prefill (prompt processing) phase, which is orders of magnitude more intensive than the decoding phase. Furthermore, the prefill phase for a request $i$ may complete in $s_i/\hat{s}$ chunks, whereas the decoding phase requires $o_i$ iterations which is way more than number of chunks. This fundamental difference in required computation and iteration count between the two phases is the primary driver of the long-tailed runtime distribution. Therefore, while _we can estimate_ the runtime for prefill and decoding batches separately, we _cannot_ assume a single, stable runtime for an "average" batch.

This concern leads me to question the choice of using a synthetic dataset primarily with a 1:2 PD (prefill-to-decode) ratio. As Appendix Figure 4 shows, as the prefill size increases (PD ratio 4:1), the batch execution time becomes more dominated by the prefill stage, and the 90th percentile runtime diverges significantly from the median. A 1:2 ratio dataset minimizes this effect, which may artificially boost the model's accuracy.

I suspect the current model is accurate for decode-dominated workloads (i.e., requests with many output tokens), as this scenario aligns better with the "stable runtime" assumption. However, for prefill-dominated workloads, the model's estimation will likely fail because the single $\bar{b}$ term cannot capture the bimodal nature of the batch runtime. This is likely why the model's worst-case error occurs on the 2:1 PD ratio dataset, suggesting the model does not generalize well.

A potential solution, which I believe would be relatively straightforward, is to split the $\bar{b}$ term. By modeling prefill batch time and decoding batch time as two distinct, estimable parameters, the subsequent derivation would likely yield a more robust model capable of handling general-purpose workloads.

**Questions:**

1. As detailed in the "Weaknesses" section, could you elaborate on the expected performance and error of the model on prefill-dominated workloads, such as LongBench dataset?

2. Could you please clarify the experimental setup described around line 376, specifically regarding how the "time interval" for the experiment was measured and used?

3. How is the $\mu_{GPU}$ (stability bound) computed? One would expect this to be an arrival rate at which the system remains stable. However, Table 1 lists $\mu_{GPU}$ as 3.387 for the 1:1 dataset, while Figure 5(b) (which seems to correspond to this experiment, though it is not explicitly linked) shows an exploding median wait time of 177 seconds at this rate. A 177-second median wait time does not represent a stable system. Could you clarify this apparent contradiction?

4. How might the model need to be adapted for different scheduling strategies, such as Shortest Job First (SJF)? I am not expecting a full derivation, but rather a high-level discussion on which components of the model would need to be revisited.

---

> ### Author Response · Authors · 2025-11-20
>
> We sincerely thank the reviewer for the thoughtful review and for recognizing the novelty and practical impact of our work.
>
> **Real Dataset Experiment.**  It is correct that real-world workloads exhibit significant variance in output lengths. To directly address this, we have conducted a new experiment using the LongBench v2 dataset, which features long, variable-length prompts and diverse, non-deterministic decode lengths (distribution figures can be found in Figure 3 on page 9).
> In the revised Section 5.2 on pages 7-8, we split the dataset 80/20 for training and testing. Using the training split, we empirically estimated the joint distribution p(s, o) and the mean batch processing time as per our methodology. The results, now presented in Table 2, show that our theoretical model achieves a high level of accuracy even on this complex, real-world data, with a gap of only 8.03% between the predicted and empirically measured service rates. This demonstrates that our framework generalizes effectively beyond simplified synthetic workloads.
>
> **High Variance Processing Time and Large P/D Ratio.**  The reviewer is correct that a high prefill-to-decode (P/D) ratio can lead to high variance in batch processing times. To investigate this rigorously, we conducted a new extreme-case experiment with a P/D ratio of 8:1. As anticipated, this resulted in a heavy-tailed distribution for batch processing time: while ~70% of batches had similar times, the remaining ~30% exhibited significant latency. This finding highlights a critical question: what is the most robust estimator for \bar{b} under high variance? In such cases, the median ignores the tail, and the mean is overly sensitive to outliers. A standard statistical solution is the trimmed mean, which removes a percentage of extreme outlier values and calculates the mean among the rest data. Applying this to our 8:1 experiment, we found: Our theory achieves a prediction error below 10%, validating our model's applicability even under high-variance conditions.
>
> **Clarification of the Time Interval.** The purpose of excluding the first and last 1,000 requests is to ensure we measure the system's performance during a stable, sustained period of operation and avoid phases that do not represent its steady-state behavior.
> Initial 1,000 requests (warm-up): At the start of the experiment, the system begins from an idle, empty state. The GPU's memory capacity is not immediately saturated, and the queueing dynamics have not reached an equilibrium. Including this initial transient phase would artificially inflate the measured service rate, as the system is not yet fully loaded.
> Final 1,000 requests (cool-down): After the arrival process stops, the system begins to drain. During this phase, there are no new requests to pack into batches, leading to under-utilized resources and a decreasing processing rate that does not reflect the system's capacity under sustained load.
>
> **The computation of $\mu_{\text{GPU}}$.** $\mu_{\text{GPU}}$ is computed in a heavily overloaded system, where the server never idles. In such a regime, we can measure the average number of prompts processed per second, which gives $\mu_{\text{GPU}}$.
> In the critical regime $\lambda = \mu$ with stochastic arrivals or departures, the system is unstable and waiting time diverges, explaining Figure 5(b). In an M/M/1 queue, $E[W] = \frac{\lambda}{\mu (\mu - \lambda)}$, which approaches infinity as $\mu - \lambda \to 0$ (Section 9.3.1 in [1]).
>
> **Different Scheduling Policy.** In fact, our models can be adapted to any work-conserving scheduling strategy—i.e., the server will accommodate the next-priority prompt whenever it is able to do so. The proofs and results remain essentially unchanged. This is because our analysis focuses on stability, which is generally not affected by the choice of work-conserving service disciplines. For most of the queueing systems, the only requirement for stability is $\rho = \lambda / \mu < 1$; see the discussion in [1]. However, if $\delta =\frac{ess\sup_{s,o\sim\mathcal{F}_{\text{in-out}}} \{ s+ o \}}{M}$is relatively large, our results still hold but are no longer tight. In this case, different scheduling policies may exhibit slightly different stability performance in practice. Moreover, for other performance metrics such as latency, different scheduling policies may lead to different outcomes.
>
> **Potential P/D Disaggregation Model.** We agree that modeling prefill and decode batch times with distinct parameters for a disaggregated architecture is a promising and important direction. This corresponds to a tandem queueing system—where requests are first processed by a prefill stage and then moved to a decode stage—which is a more advanced and realistic model.
> Deriving a stability condition for this tandem queue is a fascinating and valuable research goal.
>
> [1] Gautam, Natarajan. "Queueing theory." Operations Research and Management Science Handbook. Taylor and Francis Group (2008).

---

> > ### Comment · Reviewer_86fz · 2025-11-24
> >
> > While I appreciate the updated results and the generalization to work-conserving policies, the revised manuscript still does not adequately address my core criticism. The reliance on a simple mean estimator for both prefill and decode phases is problematic.
> >
> > Specifically, the use of a trimmed mean likely masks the issue by filtering out prefill requests (which are statistically distinct from decode batches). If $\bar{b}$ is derived from a dataset where prefills are trimmed out, it becomes an estimator for decode time only. This would naturally result in low error rates because the volume of decode tokens dwarfs prefill chunks, but it fails to validate the model for prefill phases.
> >
> > To prove the validity of this approach, could you please report the following:
> > - Results on the full 80/20 split without trimming.
> > - The count and length of prefill requests remaining in Table 4 after 5% and 10% trimming.
> > - Error rates when prefill requests are completely removed from the input.
> >
> > Unless these numbers prove otherwise, I strongly suggest splitting the runtime analysis into separate components for prefill and decode.
> >
> > On the experiment time intervals: Thank you for clarifying the experimental setup. While I agree with the necessity of excluding warm-up and drain periods, I want to confirm the exact boundaries of your data collection window.
> >
> > My understanding of your protocol is as follows:
> > 1. Warm-up: The first 1000 requests are processed but excluded from metrics.
> > 2. Measurement Window: The system runs for duration $t$.
> > 3. Drain: After time $t$, request generation stops, and the system processes remaining queued requests.
> >
> > Question: Do you include the requests that complete during the drain period in your final analysis, or is the analysis strictly limited to requests that completed within the duration $t$? Clarifying this is important because including the drain period (where load decreases) can artificially improve latency statistics.

---

> > > ### Author Response · Authors · 2025-11-25
> > >
> > > We sincerely thank the reviewer for the continued engagement and for pushing us to clarify these important methodological details.
> > >
> > > **Clarification on Trimmed Mean and Prefill/Decode Composition**
> > >
> > > The reviewer's concern that trimming might systematically remove prefill requests is understandable. We would like to clarify that this is not the case in our experiments.
> > >
> > > In the LongBench v2 experiment (8.03% GAP), we did not use any trimming. The batch processing time distribution for this real-world workload did not exhibit an extreme heavy tail, making the standard mean a sufficient estimator. The high accuracy here validates our model on a complex, realistic distribution without any post-processing.
> > >
> > > In the 8:1 P/D ratio experiment, trimming is applied to the batch processing time, not the request type. The heavy tail arises because some batches contain a larger number of prefill chunks, not because we are filtering out prefill as a category. A 5% or 10% trim only removes the most extreme outliers where the requests arrive very frequently due to the random noise in the arrival process. In this case, as vLLM prioritizes the prefill chunks, some batches contain a huge number of prefill chunks. The remaining ~90-95% of batches still contain a full mix of prefill and decode tasks (not only decode tasks). This can be validated by Figure 7, where we can see that only ~70% batches are decode-only. The remaining 30% batches are mixed by prefill and decode. Our goal was to find a robust central tendency for the overall process, and the trimmed mean—a standard statistical tool for heavy-tailed data—achieved this effectively.
> > >
> > >
> > > **Confirmation on Data Collection Window**
> > >
> > > The reviewer's understanding is correct. Our analysis is strictly limited to requests that are completed within the measurement window `T`. We explicitly exclude all requests that were completed during the drain period. This ensures that our measured service rate `μ_gpu` reflects the system's performance under sustained load and is not artificially inflated by the more efficient processing during the drain phase.
> > >
> > > **On the Foundational Nature of Our Work**
> > >
> > > We fully agree with the reviewer that a disaggregated model with separate prefill (quadratic due to attention) and decode (linear due to carrying KVs) service times is a more refined approach. However, as we noted, this leads to a tandem queueing system, whose stability analysis is fundamentally more complex and remains an open research problem.
> > >
> > > Our paper provides the essential first-step solution by solving the stability problem for the fundamental, single-stage queueing model. We believe this is a necessary and non-trivial contribution. Just as the M/M/1 queue preceded the G/G/1 queue in historical development, our work establishes the foundational theory and a closed-form stability condition upon which more complex models—like the tandem queue the reviewer proposes—can be built. It is a fundamental principle of academic progress that rigorous theoretical models are built incrementally; a single conference paper cannot capture all real-world complexities. The primary contribution of our work is to establish a foundational model that approximates LLM inference dynamics, thereby introducing a vital new research direction for the queueing theory community. We are confident that our work provides the core analytical framework to guide and inspire this future research direction.
> > >
> > > Thank you again for this rigorous discussion, which has significantly improved the clarity and strength of our manuscript.

---

> > > > ### Comment · Reviewer_86fz · 2025-11-25
> > > >
> > > > Thank you for clarifying the experimental setup; I believe incorporating this into the manuscript would significantly strengthen the paper.
> > > >
> > > > On the trimmed mean, could you share the absolute number of prefill and decode requests in the LongBench dataset before and after the training split? I am particularly interested in the prefill count within the testing dataset. Similarly, please provide the absolute counts for the 5% and 10% trimmed means in Table 4, along with the average length of those prefills.
> > > >
> > > > Finally, regarding Figure 7: I agree that 30% of requests are mixed batches and therefore a simple median or mean will not be suitable estimator (line 843). However, the use of a trimmed mean inherently truncates that tail. Since Figure 7 appears to show the distribution _before_ trimming, it would be valuable to see the distribution _after_ the trimmed mean is applied.

---

> ### Author Response · Authors · 2025-11-26
>
> We thank the reviewer for the continued engagement, which has been invaluable in strengthening our manuscript.
> **1.Experimental Setup Clarification**
> We fully agree. We have updated Section 5.2 to explicitly state that requests completed during the termination (drain) phase are excluded from all performance metrics, ensuring our measurements reflect steady-state performance under sustained load.
>
> **2. Clarification on Trimmed Mean and Dataset Composition**
> LongBench Experiment: We confirm that no trimmed mean was used for the LongBench experiment. As the CDF of batch processing times was approximately linear (without a heavy tail), the standard mean was a sufficient and accurate estimator. The dataset contains 503 requests. We used 402 requests (80%) for training. We simulate 10,000 arrival requests, where each request is randomly sampled (with replacement) from the 402 requests in the training dataset.  We use the result of this simulation to fit the joint distribution p(s,o) and estimate \bar b. The remaining 101 requests (20%) constituted the test set. In both training and testing, arrivals were simulated by randomly sampling  (with replacement) prefill lengths and their associated decode lengths.
> 8:1 P/D Ratio Experiment: The trimming is applied to batches (contain both prefills and decodes) based on their processing time, not to individual requests. Therefore, as the underlying distribution of prefill/decode ratio is 8:1, the expectation of prefill length is about 1600*8/9=1422, and the expectation of decode length is about 1600/9=178.
>
>  As requested, we have generated a new Figure 8 that displays the re-normalized CDF of batch processing times after applying the 5% and 10% trimming. This visually demonstrates how trimming removes the extreme tail while preserving the core distribution.
> We believe these clarifications and new figure directly address the reviewer's requests and hope you find this additional information satisfactory.

---

### Official Review · Reviewer_B5oJ · 2025-10-31

**Soundness:** 2
**Presentation:** 2
**Contribution:** 2
**Rating:** 4
**Confidence:** 3

**Summary:**

This work presents a queueing-theoretic framework to analyze the stability of large language model (LLM) serving under memory constraints. Based on the sequence input lengths, sequence output lengths, and average batch processing time, it constitutes a performance model when chunked prefill and first-come-first-serve (FCFS) scheduling are employed. Then, it proves the conditions (w.r.t. request arrival rate) when the system will be overloaded or stable. Numerical experiments by LLM serving show that the estimated conditions are within 10% from the ground truth.

**Strengths:**

S1. The paper is among the first ones to rigorously analyze the stability in LLM serving.

S2. The derived stability conditions

**Weaknesses:**

W1. Limited practical impact.

The analysis relies on unrealistic assumptions that the input and output lengths follow some distribution (evaluated via uniform distributions in experiments). However, in practice, the length distributions are dynamic across time (e.g., due to the request type differences across working hours and non-working hours). Such distributions cannot be known in advance, making the assessment of stability conditions infeasible.

---

W2. Over-simplification in the performance model.

The performance model makes an assumption that batch processing time remains relatively stable during the serving. However, there are two issues:
- It is known that for sequences with different lengths, the decoding speed varies substantially (e.g., see the [FlashDecoding report](https://crfm.stanford.edu/2023/10/12/flashdecoding.html) or this [vLLM issue](https://github.com/vllm-project/vllm/issues/11286)). The assumption of stable batch processing time is suitable only when the sequence lengths are within a relatively narrow range. However, when there is a mixture of long and short sequences, the assumption fails.
- If I understand correctly, a chunked prefill step is also viewed as one batch, which implies that we need to meticulously tune the chunk size in order to ensure that the time cost of one chunked prefill step is similar to that of one decoding step. However, this may not be true in practice.

Furthermore, it lacks discussion/comparison with the performance models used in existing LLM simulators based on profiling and cost modelling (e.g., Vidur [1] and Scratchpad [2]).

[1] Vidur: A Large-Scale Simulation Framework for LLM Inference. \
[2] Scratchpad. https://github.com/eth-easl/Scratchpad/tree/main/tools/simulator

---

W3. The experiments are conducted with a simplified setup (synthetic, constant workloads). And several key metrics like percentile latency and service level objective (SLO) are not considered.

**Questions:**

Please refer to the weaknesses.

---

> ### Author Response · Authors · 2025-11-20
>
> We sincerely thank the reviewer for the thoughtful review and for recognizing the novelty of our work.
>
> **Unknown distribution and limited impact.**  We agree that in practice, workload distributions can be dynamic. However, this does not render our stability condition infeasible; rather, it defines its operational context. Our framework provides the fundamental relationship $\mu = f(M, \mathcal{F}\_{\text{in-out}}, A,\bar{b})$. In a production environment, system operators can leverage well-established techniques for time-series forecasting and distribution estimation—where workloads often exhibit strong daily or weekly patterns (e.g., distinguishing peak business hours from off-peak periods)—to maintain a running estimate of $\mathcal{F}_{\text{in-out}}$ for a given time window. Our stability condition then becomes a powerful tool for dynamic provisioning: by periodically re-calculating $\mu$ with the updated distribution, an autoscaling system can add or remove GPU replicas to maintain $\lambda<\mu$ in near-real-time, precisely as suggested in our Section 4. Thus, our work provides the critical core model upon which adaptive, production-grade scaling controllers can be built.
>
> Regarding academic impact, as the first queueing-theoretic model to incorporate KV cache memory constraints, our primary contribution is to establish a rigorous foundational theory for a new and critical domain. It is precisely because the real-world system is complex (with non-stationarity, variable service times, etc.) that we must start with a tractable baseline model to derive closed-form stability conditions. Our empirically validated theory also provides this essential foundation.
>
> **Non-constant batch processing time.**   We acknowledge that in reality, processing time varies with batch composition.
> However, for the purpose of stability analysis, we believe that using a constant batch size is a good approximation, as supported by our numerical examples: despite this simplification, the predicted stability condition is highly accurate, with errors typically below 10%. Furthermore, for other scenarios—including those with a non-fixed prefill/decode ratio that lead to more variable batch processing times, as shown in the updated paper—our results remain fairly accurate (gap of $\approx 8\%$ on a real-world dataset and <10% in the P:D = 8:1 case).
>
> We fully agree that incorporating more sophisticated service time models (e.g., under prefill-decode (PD) disaggregation, prefill time is quadratic in token count due to attention, decode time is linear due to carrying time of KV) is a valuable and logical next step, and we have outlined this explicitly in our "Future Work" section (Section 6). Our current work provides the essential theoretical backbone upon which such more granular, performance-oriented analyses can be built.
>
> **Experiments with Vidur.** We argue that they serve a different and complementary purpose to our analytical model, especially for determining stability conditions.
> Using such simulators to find the stability limit is theoretically possible but practically infeasible for two key reasons:
>  *High Variance and Computational Cost*: Simulators require specific arrival traces. To estimate a stability condition—a fundamental property of the system and workload distribution—one must run a massive number of Monte Carlo simulations over different trace samples to achieve statistical confidence. This process is computationally prohibitive and time-consuming.
> *Inefficient Search for the Critical Point*: Finding the precise stability boundary (the minimum number of GPUs required) via simulation requires an iterative search (e.g., binary search over GPU counts). Each iteration in this search requires a long-duration simulation to distinguish stable from unstable regimes, compounding the computational burden.
>
> In contrast, our analytical model provides an instant, closed-form expression for the critical service rate $\mu$. By simply computing $\mu = f(M, F_{in-out}, A, \bar{b})$, a system designer can immediately determine the necessary resources. Our experiments demonstrate this method is both highly accurate (within 10%) and requires negligible computation. Therefore, our work provides a foundational theory-first approach that is uniquely suited for rapid capacity planning and theoretical analysis, complementing the simulation-first approach used for detailed performance evaluation of specific, fixed configurations.
>
> **Other metrics.**  In this work, we focus primarily on stability, which concerns whether the system can eventually process all arriving prompts. In queueing systems, the stability condition is fundamental: while different scheduling policies may lead to drastically different SLOs, the stability requirement itself is typically similar—roughly $\lambda < \mu$. Therefore, we view our work as a fundamental and necessary first step. In future work, we will study scheduling policies that optimize other SLOs.

---

> > ### Comment · Reviewer_B5oJ · 2025-11-26
> >
> > Can you refer to some notable examples of "well-established techniques for time-series forecasting and distribution estimation"?
> >
> > If I understand correctly, the proposed queueing-theoretic framework requires knowledge about the length distribution, rather than summary statistics like mean lengths. Do you need to enforce several candidate priors about the distribution within a time window? Otherwise, do you need to estimate percentiles of lengths for each time window?
> >
> > More importantly, my concern in W1 is about the feasibility/practicality. If complex estimation methods are necessary and the estimation accuracy matters, I can hardly say the feasibility/practicality issue can be ignored.
> >
> > ---
> >
> > As noted in the rebuttal, it is practically infeasible to find the stability limit using simulators that are more accurate but more complex than the performance model in this work. This raises another question that whether it implies that simplifications in the performance model is necessary to find the stability limit? If yes, can the simplified performance model well represents real-world scenarios?
> >
> > Besides, I didn't request experiments using Vidur/Scratchpad. However, it would help to compare the accuracy of performance model with Vidur/Scratchpad.
> >
> > ---
> >
> > It would help if you can highlight which parts are updated in the manuscript.

---

> > > ### Author Response · Authors · 2025-11-27
> > >
> > > **On Time-series Forecasting and Distribution Estimation**.
> > > Time-series forecasting is a well-established field, with notable methods including ARIMA [1], exponential smoothing, and feature-based regression with ARMA errors [3]. In data-center applications, we recommend the feature-based ARMA approach, as the arrival process exhibits clear nonstationarities such as day-of-week and time-of-day patterns. These provide a rich set of covariates that can be leveraged effectively.
> > > Specifically, given features $x_t$​, we model the observations a
> > > $$Y_t=x_t^\top \beta + u_t,$$
> > > where the residual process $u_t$ follows an ARMA model.
> > >
> > > [1] Box, George EP, et al. Time series analysis: forecasting and control. John Wiley & Sons, 2015.
> > > [2] Hyndman, Rob J., and George Athanasopoulos. Forecasting: principles and practice. OTexts, 2018.
> > > [3] Kedem, Benjamin, and Konstantinos Fokianos. Regression models for time series analysis. John Wiley & Sons, 2005.
> > >
> > > **On Distribution Knowledge vs. Summary Statistics**.
> > > The reviewer is correct that our framework utilizes the full joint distribution of prefill and decode lengths, not just summary statistics. While this increases the complexity of the derivation, it is a fundamental strength. Stability is determined by the tail behavior and the complex interaction between prompt and output lengths. Two distributions with the same mean (e.g., geometric vs. uniform) can lead to significantly different stability boundaries, and our model is the first to capture this crucial effect.
> > >
> > > **On Practical Feasibility of Estimation**.
> > > We agree that estimation is key to practicality. Fortunately, in production environments, workload patterns are highly predictable over daily or weekly cycles. A standard and highly feasible approach in real industry is to use historical data from a corresponding time window (e.g., data from 9-10am over the previous month) to accurately estimate the joint distribution for the current 9-10am window. One can also leverage the time series approach mentioned above. Our LongBench v2 experiment directly validates this methodology: we used 80% of the dataset (the "historical" data) to learn the distribution and then achieved high accuracy on the unseen 20% test set (the "current day's" data). For the experiment, we never use the distribution information of test dataset. This demonstrates the framework's robustness to distributional estimation and its direct practical feasibility.
> > >
> > > **Necessity of our Performance Model**.
> > > We argue that both robust modeling of reality and tractability are core principles in LLM inference queueing model design. Extremely detailed performance models may be more accurate in certain scenarios, but they often fail to generalize and thus compromise robustness, especially as LLM inference algorithms, model infrastructures, and hardware evolve rapidly. Moreover, such complex models inevitably lose tractability, making it difficult to extract useful insights from the theoretical analysis. In contrast, our simplified model captures the key characteristics of LLM inference and provides a close approximation to practice, as demonstrated by our extensive numerical experiments.
> > >
> > > **Highlight Updates in the Manuscript**
> > > Thanks for the suggestions. We have highlighted major changes in blue in the updated manuscript. Specifically:
> > > 1. We conducted additional numerical experiments, including those on real-world datasets (LongBench v2) in Section 5.2 and the case (P:D = 8:1) in Appendix C.
> > >
> > > 2. We generalized our results to any work-conserving policy (Page 5).
> > >
> > > 3. We added a discussion of a relevant paper (Page 3).

---

### Official Review · Reviewer_y4eS · 2025-11-01

**Soundness:** 3
**Presentation:** 4
**Contribution:** 3
**Rating:** 6
**Confidence:** 2

**Summary:**

This paper presents a queueing theory framework for LLM serving that, for the first time, accounts for GPU memory constraints in addition to computation. It provides a tool for analyzing LLM serving systems, and its accuracy is validated through actual measurements, showing deviations within 10%.

**Strengths:**

- This is the first work to consider KV cache constraints in the queueing-theoretic analysis of LLM serving systems.
- The theoretical predictions closely match actual measurements obtained with vLLM.

**Weaknesses:**

- The assumption about the essential supremum (ess sup {s + o} << M) may be too strong. Section 5.1 states that $M$ is 131,000 for the 8B model, which would be smaller for larger models. Considering that context lengths can typically reach tens of thousands of tokens [1, 2], this assumption may need to be relaxed for a more realistic analysis.
- The systems considered in this paper may be overly simplified and not fully representative of real-world LLM serving environments. See the questions below for more details.

**Questions:**

The theoretical framework could be improved by modeling more realistic settings by considering the following points:

- What if the KV cache is not released after a request finishes? For chatbot or agent applications, it is common to store KV caches in some way for future interactions [3, 4, 5].
- What if the per-iteration latency ($\bar{b}$) exhibits high variance? I think this happens when the arrival patterns of requests are skewed.
- What if more realistic request length distributions are considered, instead of fixed PD ratios?

Please note that I am not an expert in queueing theory, so I am not entirely certain whether the above points are necessary for the paper's acceptance.

### References

- [1] https://github.com/Azure/AzurePublicDataset
- [2] https://arxiv.org/abs/2407.00079
- [3] https://arxiv.org/abs/2311.04934
- [4] https://arxiv.org/abs/2408.12757
- [5] https://arxiv.org/abs/2510.09665

---

> ### Author Response · Authors · 2025-11-20
>
> We sincerely thank the reviewer for the thoughtful review and for recognizing the novelty and practical impact of our work.
>
> **Small workload assumption.** We thank the reviewer for this critical question regarding the `ess sup` assumption.
> First, we clarify that `M` represents the KV cache capacity, which is the total GPU memory *minus* the space for model weights and system overhead. When deploying longer requests (s+o larger) and larger models, practitioners provision hardware (e.g., using H100s or multi-GPU configs) specifically to ensure a sufficiently large `M` to enable cost-effective, high-throughput inference through batching. In our experiments, we use context lengths up to 2133*3 tokens and `M=131,000`. In this case, $\delta \approx 2133 \times 3/131000 = 0.048  \ll 1$, confirming its practical utility for common serving scenarios focused on high throughput.
>
> Second, from a theoretical perspective, Theorems 1 and 2 hold for any $\delta \in [0,1]$. The contribution of Theorem 2 is to show that a specific policy (work-conserving with lookahead) can achieve a rate close to this limit when $\delta$ is small.
>
>
> **Realistic system (i) KV cache is not released.**  The reviewer is absolutely correct, and we agree that this is a vital scenario. Importantly, our queueing-theoretic framework can be naturally extended to model this.
>
> The key insight is that our stability condition is derived from the total "memory-area" demand of each request. For a request with prompt length `s`, output length `o`, and a persistence time `p`, its total memory-area becomes `g(s, o, p) = g_original(s, o) + p * (s+o)`. The first term captures the area during active processing, and the second term captures the area during idle persistence. The core of our Lyapunov argument—comparing the arrival rate of memory-area to the system's capacity `M/b̄`—remains unchanged.
> Therefore, the stability condition `λ < M / (b̄ E[g(s, o, p)])` still holds in this generalized form. This demonstrates the flexibility of our memory-area principle. For the specific case studied in our paper, we assumed `p=0` (immediate release after generation), which models a common API-serving pattern. Explicitly incorporating a distribution for `p` to analyze chatbot workloads is a direct and tractable next step, which we can now formally characterize thanks to this discussion.
>
> **Realistic system (ii) $\bar{b}$ has high variance.** The reviewer is correct that skewed request patterns, such as those with a high prefill-to-decode (P/D) ratio, can lead to high variance in batch processing times.
>
> To investigate this rigorously, we conducted a new extreme-case experiment with a P/D ratio of 8:1. As anticipated, this resulted in a heavy-tailed distribution for batch processing time: while ~70% of batches had similar times, the remaining ~30% exhibited significant latency. This finding highlights a critical question: what is the most robust estimator for \bar{b} under high variance? In such cases, the median ignores the tail, and the mean is overly sensitive to outliers. A standard statistical solution is the trimmed mean, which removes a percentage of extreme outlier values and calculates the mean among the rest data. Applying this to our 8:1 experiment, we found:
>  Using a 5% trimmed mean, $\bar{b} = 0.093$ yielded $\mu_{theory} = 4.977$, a 9.0% gap from the empirical $\mu_{gpu} = 5.47$.
> Using a 10% trimmed mean, $\bar{b} = 0.078$  yielded $\mu_{theory} = 5.862$, a 7.2% gap.
> Both robust estimators achieve a prediction error below 10%, validating our model's applicability even under high-variance conditions. We sincerely thank the reviewer for this insight, which underscores that selecting an appropriate statistical estimator for $\bar{b}$ is key to handling real-world skew.
>
> **Realistic system (iii) non-fixed ratio of prefill/decode.**   It is correct that real-world workloads exhibit significant variance in output lengths. To directly address this, we have conducted a new experiment using the LongBench v2 dataset (as suggested by Reviewer 4), which features long, variable-length prompts and diverse, non-deterministic decode lengths (distribution figures can be found in Figure 3 on page 9). The data is publicly accessible at https://longbench2.github.io/.
> In the revised Section 5.2 on page 7-8, we split the dataset 80/20 for training and testing. Using the training split, we empirically estimated the joint distribution p(s, o) and the mean batch processing time as per our methodology. The results, now presented in Table 2, show that our theoretical model achieves a high level of accuracy even on this complex, real-world data, with a gap of only 8.03% between the predicted and empirically measured service rates.
> This demonstrates that our framework, which explicitly accounts for the full joint distribution of prefill and decode lengths (including their variance and dependency), generalizes effectively beyond simplified synthetic workloads.

---

> > ### Comment · Reviewer_y4eS · 2025-11-27
> >
> > Thanks for your response. Please be sure to include those analyses in the modified manuscript.
> >
> > **Small workload assumption**.
> > Could you provide more details about what happens to Theorem 2 if the assumption $\delta \ll 1$ were relaxed? Would the work-conserving-with-lookahead policy still achieve a rate close to the theoretical limit, or does the proof critically depend on $\delta$ being asymptotically small?
> >
> > Also, could you discuss more on whether the practical value of $\delta$ is small enough? It is around $0.05$ in your case and can be higher depending on the hardware or workload configurations. Since the proposed theoretical model's error is on the order of ~10%, I wonder whether the $\delta$ is something we can ignore.
> >
> > **Realistic system (ii)**.
> > P/D ratio of 8:1 is good but not very extreme. I want to know if a similar property holds for more extreme cases where prefill length is orders of magnitude larger than decode length (or vice versa).

---

> > > ### Author Response · Authors · 2025-11-28
> > >
> > > We thank the reviewer for the continued engagement, which has been invaluable in strengthening our manuscript.
> > >
> > > **Small workload assumption**
> > > Thanks for raising this question. First, the conclusions of Theorem 2 continue to hold even when (\delta) is large: if
> > > $
> > >  \lambda < \mu(1-\delta),
> > >  $
> > >  then under any work-conserving scheduling and batching policy, the system remains stable for all (\delta < 1).
> > > Second, when $\lambda \in [\mu(1-\delta), \mu)$, different policies may exhibit different stability behavior. For example, we expect Shortest Job First (SJF) to perform better in the sense that it can maintain stability even when $\lambda$ is relatively large compared to $\mu(1-\delta)$.
> > > Finally, in practice, when deploying longer requests (larger (s + o)) and larger models, practitioners typically provision more capable hardware (e.g., H100s or multi-GPU setups) to ensure a sufficiently large (M), thereby enabling cost-effective, high-throughput inference via batching. For these reasons, we view the regime $\delta$ < 10% as practically relevant. Since model-estimation error is often on the order of roughly 10%, the $\delta$-term can be usually ignored.
> > >
> > >
> > > **Realistic system (ii)**
> > > In the new Section 5.3, we present experiments on the LongBench v2 dataset, which represents an extreme, real-world workload where prefill lengths dominate decode lengths by orders of magnitude. As detailed in the new Figure 3 (page 9), the statistics for this dataset are:
> > > >Prefill: Median = 11,863 tokens, Mean = 15,274 tokens.
> > > Decode: Median = 13 tokens, Mean = 87 tokens.
> > >
> > > This yields a median P/D ratio of 11,863 / 13 ≈ 912, a scenario where prefill overwhelmingly dominates. Crucially, in this regime, the batch processing times did not exhibit a heavy tail, as most of the batches consist of many equally sized prefill chunks. Consequently, as shown in Table 2 in our updated manuscript, our theoretical model maintains high accuracy, with a gap of only 8.03% compared to the real vLLM measurements.
> > > This result demonstrates that our framework is effective not only for synthetic extremes (like the 8:1 ratio) but also for real-world, highly skewed workloads where prefill lengths are several orders of magnitude larger than decode lengths.

---

> > > > ### Comment · Reviewer_y4eS · 2025-11-28
> > > >
> > > > Thanks again for the clarification.
> > > >
> > > > > For these reasons, we view the regime $\delta$ < 10% as practically relevant. Since model-estimation error is often on the order of roughly 10%, the $\delta$-term can be usually ignored.
> > > >
> > > > I still need some more clarification about this point.
> > > > Specifically, is the prediction error empirically correlated with the magnitude of $\delta$? Does the error gap shrink in experiments where $\delta$ is smaller (closer to the theoretical fluid limit), or does the ~10% error persist as a noise regardless of $\delta$? I want to understand whether the approximation that ignores $\delta$ is a significant source of the current prediction error, or whether the error is dominated by other system noise.

---

> > > > > ### Author Response · Authors · 2025-11-28
> > > > >
> > > > > Thanks for your excellent question. We do not observe any correlation between the model-prediction error and the magnitude of $\delta$. In our synthetic experiments, for the PD ratio of 1:2, we have $\delta \approx 2133 \times 3 / 131000 \approx 0.048$, and for the PD ratio of 8:1, we have $\delta \approx 3200 / 131000 \approx 0.024$. For the LongBench dataset, the value of $\delta$ is even more extreme. Although these $\delta$ values differ substantially, the model-prediction errors remain similar, all in the range of 5%–10%. Therefore, we believe that the dominant sources of error arise from other factors.

---

### Official Review · Reviewer_Wq99 · 2025-11-01

**Soundness:** 3
**Presentation:** 3
**Contribution:** 3
**Rating:** 6
**Confidence:** 3

**Summary:**

This paper studies LLM inference as a queueing system where both compute and memory are limiting resources. The authors model prompt arrivals as a Poisson process with per-request prompt length $s$ and decode length $o$, and formalize how each request consumes KV cache over time during the prefill and decode phases. They define a cumulative memory demand function $g(s,o)$ for each request and use it to derive a closed-form expression for an effective processing rate $\mu$ as a function of the memory capacity of the GPU, the joint distribution of prompt/response lengths, the scheduler, and per-batch serving time. The main theoretical contributions are stability conditions for the LLM serving systems. The paper then validates these stability predictions on real A100 hardware running Llama-3 under vLLM-style continuous batching and chunked prefill, across several prompt/decode length ratios. The predicted stable service rate $\mu$ matches the measured throughput within 10%, suggesting this analysis can be used for capacity planning.

**Strengths:**

- The authors propose a novel perspective for the stability analysis of LLM inference systems considering memory constraints. Prior work largely considered compute/throughput as the bottleneck; here, KV cache usage is more explicitly modeled as a limiting resource over the lifetime of a request.
- I think that the theoretical results are simplistic and actionable; they provide basic rules for practitioners while deploying LLM inference systems.
- The authors provide comprehensive numerical results that cover single and multi-GPU settings, and they show that their estimates work.

**Weaknesses:**

- The model treats per-batch processing time as constant, justified empirically by showing 80% of the batches have identical processing times. I am not sure how the heterogeneity of the batches (what percentage is prefill vs decoding) affects the batch processing time for any given batch.
- The experiments force a fixed prefill/decode ratio by setting decode length to a deterministic function of the prompt length. That is convenient for control but not representative of real chat workloads, where decode length can vary and can exceed prompt length by orders of magnitude. It would also be interesting to see a robustness analysis or extension when the output lengths are estimated using the prompt information.

**Questions:**

- The service-rate definition Eq. (1) uses the total lifetime memory area $g(s,o)$, but the GPU memory constraints need to be satisfied instantaneously during the inference. Could the authors clarify how this aggregate memory-time simplification captures momentary memory saturation, and under what conditions the area-based estimate may under or over-estimate the real service-rate?

- The related work section could benefit from citing [1], which also analyzes queueing-based throughput models for LLM inference. This would help clarify how the present formulation differs in assumptions or theoretical guarantees.
[1] Guldogan, Ozgur, et al. "Multi-bin batching for increasing LLM inference throughput." arXiv preprint arXiv:2412.04504 (2024).

---

> ### Author Response · Authors · 2025-11-20
>
> We sincerely thank the reviewer for the thoughtful review and for recognizing the novelty and practical impact of our work. In response to the valuable questions and weakness points raised, we have incorporated clarifications and additional discussions into the manuscript. Below, we address each point in detail.
>
> **Fixed per-batch processing time.**. We thank the reviewer for this important question regarding the constant batch processing time assumption. We acknowledge that in reality, processing time varies with batch composition; a batch with more prefill tokens requires more computation due to the quadratic attention cost, while decode-heavy batches are typically faster and linear.
>
> However, for the purpose of stability analysis, we believe that using a constant batch size is a good approximation, as supported by our numerical examples: despite this simplification, the predicted stability condition is highly accurate, with errors typically below 10%. Furthermore, for other scenarios—including those with a non-fixed prefill/decode ratio that lead to more variable batch processing times, as shown in the updated paper—our results remain fairly accurate (gap of $\approx 8\%$ on a real-world dataset and <10% in the P:D = 8:1 case).
>
> We fully agree that incorporating more sophisticated service time models (e.g., under prefill-decode (PD) disaggregation, prefill time is quadratic in token count due to attention, decode time is linear due to carrying time of KV) is a valuable and logical next step, and we have outlined this explicitly in our "Future Work" section (Section 6). Our current work provides the essential theoretical backbone upon which such more granular, performance-oriented analyses can be built.
> Experiments for non-fixed ratio of prefill/decode. We thank the reviewer for this insightful observation regarding the representativeness of fixed P/D ratios. It is correct that real-world workloads exhibit significant variance in output lengths. To directly address this, we have conducted a new experiment using the LongBench v2 dataset (as suggested by Reviewer 4), which features long, variable-length prompts and diverse, non-deterministic decode lengths (distribution figures are presented in Figure 3 on page 9). The data is publicly accessible at https://longbench2.github.io/.
> In the revised Section 5.2 on page 7-8, we split the dataset 80/20 for training and testing. Using the training split, we empirically estimated the joint distribution p(s, o) and the mean batch processing time as per our methodology. The results, now presented in Table 2, show that our theoretical model achieves a high level of accuracy even on this complex, real-world data, with a gap of only 8.03% between the predicted ($\mu_{\text{theory}}=0.561$)  and empirically measured ( $\mu_{\text{gpu}}=0.610$) service rates.
> This demonstrates that our framework, which explicitly accounts for the full joint distribution of prefill and decode lengths (including their variance and dependency), generalizes effectively beyond simplified synthetic workloads. We are confident that this addition directly addresses the reviewer's concern and substantially improves the paper's practical relevance. We sincerely thank the reviewer for this valuable advice.
>
> **Clarify Eq 1.**  We thank the reviewer for this insightful question. The use of the lifetime memory area $g(s, o)$ is indeed an aggregate measure, but its connection to instantaneous constraints is rigorously established in our Lyapunov stability proof (Theorem 2, Appendix A). The core insight is that when the backlog is large, the FCFS policy with memory lookahead can continuously form batches that keep the GPU's memory utilization near its capacity $M$ without risk of overflow. The small slack $\delta$ is precisely the headroom needed to accommodate the largest possible single-request memory increment. In this saturated regime, the system 'erases' memory area at a nearly constant rate of $ M/\bar{b} $ per second. Therefore, the area-based rate $ \mu $ becomes the limiting factor. The estimate may be conservative (under-estimate the real service rate) in non-saturated, low-queue regimes where instantaneous packing is less efficient, but it provides a provable lower bound for stability, which is our primary goal for provisioning.
>
> **Citation of Guldogan, Ozgur, et al (2024):** We thank the reviewer for pointing out this interesting and relevant paper. It also studies LLM inference from a queueing perspective. However, it does not explicitly model the KV cache. We have added a citation to this work and incorporated a discussion of it in the revised manuscript.

---

> > ### Comment · Reviewer_Wq99 · 2025-11-26
> >
> > Thanks to the authors' detailed and thoughtful responses.
> >
> > First, thank you for the extended results regarding the heterogeneity of batch processing times. The additional experiments, especially those involving non-fixed prefill/decode ratios and a real dataset (LongBench v2), strengthen the practical relevance of the work. I agree with the authors that the constant batch-processing time assumption is a reasonable first step for the stability analysis, and that modeling heterogeneous prefill/decode dynamics is a possible direction for future work.
> >
> > Regarding the clarification of Eq. (1) and the stability proof; my understanding is that the proof shows that the total outstanding memory decreases whenever the demand exceeds a certain threshold, ensuring that even in heavy-traffic regimes, there is always a small request that can be included in the next batch processing. This guarantees that the memory erasure speed remains faster than the incoming demand growth, establishing stability. If I am not mistaken, I did notice a small typo at the end of the proof of theorem 2, the last inequality appears to be missing a $\bar{b}$ term.
> >
> > Overall, I appreciate the authors' explanations, and I will maintain my positive rating of the paper.

---

> > > ### Author Response · Authors · 2025-11-26
> > >
> > > Your understanding is absolutely correct. Thank you for pointing out the typo at the end of the proof of Theorem 2. We have corrected it in the updated manuscript.

---

### Author Response · Authors · 2025-11-20

We sincerely thank all the reviewers for their valuable comments. We have updated our paper accordingly. Specifically:
1. We conducted additional numerical experiments, including those on real-world datasets (LongBench v2) and the P:D = 8:1 case, and we show that our results remain highly accurate in these regimes.


2. We generalized our results to any work-conserving policy.


3. We added several missing references.

We would like to conclude by emphasizing a key point: as the reviewer acknowledges, our work is the first to derive a theoretical stability condition for LLM serving under KV cache constraints.

Establishing a rigorous queueing-theoretic foundation for a system as complex as LLM inference is an incremental process, much like the historical development of queueing theory itself, which progressed from simple models (M/M/1) to increasingly general ones (M/G/1, G/G/1). Similarly, practical systems like vLLM were not fully optimized in their first version but served as vital foundations for community-driven improvement.

Therefore, we position our paper as this essential first step—a foundational model that provides the first closed-form stability condition. It opens the door for future work to relax assumptions and model more advanced serving paradigms. We hope our work encourages more researchers in queueing theory to engage with the significant challenges in modern LLM serving.

---

### Author Response · Authors · 2025-12-01
**Summary of our contributions and improvements made during the rebuttal process**

Dear Reviewers and the AC,

We are grateful for the opportunity to highlight our contributions and the significant improvements made during the rebuttal process.
As noted by all reviewers, our paper presents the **first queueing-theoretic framework that derives a rigorous, closed-form stability condition for LLM inference under KV-cache memory constraints.** This foundational contribution bridges queueing theory and large-scale AI systems, providing the mathematical basis for addressing the central deployment problem in LLM inference. We validate the model and theory against the **real-world vLLM system using both synthetic and real datasets**, where it predicts the stability threshold with high accuracy (typically within 10%). The resulting stability condition directly informs capacity planning—determining the minimum number of GPUs needed to support a stream of requests—and our theoretical framework serves as a building block for future research on LLM inference.


## Rebuttal Summary:

In response to reviewer feedback, we have substantially strengthened the paper's empirical validation and practical relevance:
1. **Robustness to Extreme P/D Ratio:** Reviewers requested validation under more extreme Prefill/Decode (P/D) ratios. We conducted two new experiments:

    * Appendix C: A synthetic workload with an 8:1 P/D ratio, where our model maintains a <9% error.

    * Section 5.2: An experiment on the real-world LongBench v2 dataset. This represents an extreme case where the median P/D ratio is ~912:1 (prefill median=11,863 tokens, decode median=13 tokens). Even here, our model's error is only 8.03%, demonstrating its applicability to highly skewed, real-world scenarios.

2. **Modeling Assumptions and Generality:** We addressed questions about the constant batch processing time assumption. We showed that for heavy-tailed distributions, a robust statistical estimator (trimmed mean) for the processing time preserves high accuracy.

3. **Practical Feasibility:** We clarified that our framework's reliance on the joint length distribution is a strength for accuracy and is highly practical. A standard and highly feasible approach in real industry is to use historical data from a corresponding time window (e.g., data from 9-10am over the previous month) to accurately estimate the joint distribution for the current window.

4. **Model/Theory Clarification.** We generalize our theoretical framework to all work-conserving scheduling policies. Furthermore, we clarify that even when the load factor $\delta$ is close to one, where our theoretical bounds are not sufficiently tight, the resulting stability conditions still provide accurate predictions.

In conclusion, we position this paper as the **essential first step in a new research direction.**  We hope our work encourages more researchers to engage with these significant challenges at the intersection of theory and systems.

---

### Meta-Review · Area_Chair_KrAj · 2026-01-13

**Summary:**

Reviewers broadly agreed that the paper makes a novel and timely theoretical contribution by being the first to derive a closed-form, queueing-theoretic stability condition for LLM inference that explicitly accounts for KV-cache memory constraints. Multiple reviewers highlighted the clarity of the core idea, the usefulness of the stability condition for capacity planning, and the strong empirical validation, with predicted stability thresholds matching real GPU measurements within ~10%.

The main concerns centered on the simplifying assumptions required for tractability—most notably (i) the assumption of a constant (or effectively constant) batch processing time despite known heterogeneity between prefill and decode phases, (ii) assumptions on workload size relative to memory capacity (small-δ regime), and (iii) the practicality of estimating full joint prompt/output length distributions in non-stationary, real-world workloads. Some reviewers also questioned whether the experimental workloads were sufficiently representative and whether the model should explicitly disaggregate prefill and decode service times.

**Reviewer Concerns:**

Concerns Largely Addressed by the Rebuttal

- Representativeness of workloads: The addition of experiments on LongBench v2 (with highly skewed and variable prefill/decode lengths, including extreme P/D ratios) directly addressed concerns about synthetic or fixed-ratio workloads.

- Batch-time variability and heavy tails: Reviewers’ concerns about non-constant batch processing time were mitigated through new experiments (e.g., 8:1 P/D ratio), analysis of heavy-tailed behavior, and the use of robust estimators (trimmed mean), showing prediction errors remain within ~10%.

- Generality of the theory: The authors clarified and extended results to all work-conserving scheduling policies, addressing questions about FCFS-specific assumptions.

- Interpretation of the memory-area (area-based) stability condition: Detailed clarifications tied the aggregate memory-area argument to instantaneous memory constraints via the Lyapunov proof, resolving conceptual confusion for multiple reviewers.

- Practical feasibility of distribution estimation: The rebuttal convincingly argued (and empirically demonstrated via train/test splits) that estimating joint length distributions from historical data windows is realistic in production settings.

Concerns That Remain Partially or Fully Outstanding

- Prefill/decode disaggregation: Several reviewers (especially Reviewer 86fz) remain unconvinced that a single batch-time parameter is fully adequate, and continue to argue for an explicit two-stage (prefill vs. decode) or tandem-queue model.

- Model tightness outside the small-δ regime: While the authors argue that the regime δ < ~10% is practically relevant and that prediction error is not empirically correlated with δ, some uncertainty remains about tightness guarantees when δ is larger.

- Scope of practical impact: One reviewer (B5oJ) still views the reliance on full distribution estimation and simplified modeling as limiting near-term deployment impact, despite the authors’ arguments.

**Reviewer Scores:**

Reviewer Wq99 (initial rating: 6)
Likely to maintain a 6, given their explicit statement that the added experiments and clarifications strengthened the paper and that they would maintain a positive rating.

Reviewer y4eS (initial rating: 6)
Likely to maintain. The reviewer’s main theoretical concerns were addressed through extended discussion and experiments, and later comments suggest improved confidence, though with some lingering curiosity rather than objection.

Reviewer B5oJ (initial rating: 4)
Likely to maintain (4). While still skeptical about feasibility and simplifications, many practical concerns (distribution estimation, comparison to simulators, scope of claims) were addressed in detail.

Reviewer 86fz (initial rating: 4)
Likely to remain at 4. Although the reviewer continued to push on prefill/decode disaggregation and trimmed-mean interpretation, they acknowledged that the clarifications and added experiments strengthened the manuscript.

---

### Decision · Program_Chairs · 2026-01-26

Reject